

**Comment on:**
**Macroscopic water vapor diffusion is not enhanced in snow**
**Andrew C. Hansen**
Professor Emeritus, University of Wyoming, Laramie, WY 82071 USA
*Correspondence to:* A.C. Hansen (hansen@uwyo.edu)
December 2021
**Abstract**
The central thesis of the authors' paper is that macroscopic water vapor diffusion is not enhanced
in snow compared to diffusion through humid air alone. Further, mass diffusion occurs entirely as the
result of water vapor diffusion in the humid air at the microscale and the ice phase has no effect other than
occupying volume where diffusion cannot occur. The foundation of their conclusion relies on the premise
that the synchronous sublimation and deposition of water vapor across ice grains, known as hand-to-hand
water vapor transport, does not lead to enhanced mass diffusion. We use a layered microstructure to
rigorously show that diffusion is enhanced at all ice volume fractions compared to diffusion through
humid air alone, and further, the hand-to-hand model of diffusion correctly predicts this diffusion
enhancement.
The authors attempt to dismiss the concept of enhanced mass transfer resulting from hand-to-
hand water vapor transport by arguing that there is a "counterflux" of water vapor in the form of
downward motion of ice. While the ice phase *appears* to be propagating downward, all continuum
material points of water (either vapor or ice) are moving upward (counter to the temperature gradient)
with a monotonically increasing (nonnegative) motion. Specifically, material points of water in vapor
form are diffusing upward through the humid air while material points of water in the form of ice are at
zero velocity while locked in the ice phase. Material points of water never exhibit downward motion,
despite the ice phase *appearance* of downward motion. *Since the motion of all material points of water is*
*monotonically increasing for all time, there is no counterflux of mass due to downward motion of the ice*
*and such apparent motion is a mirage in the context of mass transfer.*
This paper presents a rigorous fluid mechanics control volume analysis of mass transfer to
demonstrate that the hand-to-hand model of diffusion produces the correct diffusion coefficient for the
layered microstructure. Moreover, the control volume analysis shows why the authors' approach of
volume averaging the microscale diffusion coefficient does not capture the complete water vapor mass
transport and therefore does not produce the correct macroscale diffusion coefficient.
An entirely fresh perspective on the role of the ice phase in mass diffusion is also presented. In
particular, an analysis showing diffusion enhancement is developed without resorting to the hand-to-hand
diffusion analogy. In brief, rather than looking at the ice as blocking microscale diffusion, the ice phase
should be viewed as a reservoir of water, containing vast amounts of water vapor, ready to be released in
the diffusion process.
In conclusion, mass diffusion in a layered microstructure is enhanced at all ice volume fractions
compared to diffusion through humid air as a pure substance. The mechanism producing this enhanced
diffusion is also on full display in snow under strong temperature gradients. Hence, it is entirely possible,
indeed probable, that macroscopic water vapor diffusion is enhanced in snow compared to diffusion in
humid air as a pure substance.





## 1. Introduction

Heat and mass transfer at the macroscale of a snow cover is a complex phenomenon, even under the simplest of conditions. The challenges in modeling thermophysical processes in snow stem from the fact that snow is a phase changing mixture of ice and humid air. Under a macroscale temperature gradient, the transport properties for snow are influenced by water vapor diffusion. Diffusion is, in turn, influenced by several microscale factors including elevated temperature gradients in the humid air as well as the complex 3D topology of the ice phase. However, without question, the most vexing aspect of modeling diffusion is the condensation and sublimation of water molecules resulting in "hand-to-hand" water vapor transport as famously described by Yosida (1955).

Figure 1 shows two forms of water vapor transport in snow under the influence of a macroscale temperature gradient. Some water vapor molecules follow paths around ice grains while others undergo sublimation and condensation, resulting in the hand-to-hand vapor transport described by Yosida. While the existence of hand-to-hand water vapor transport is well known for some 60+ years, there remains controversary surrounding the relation of this mass transfer mechanism to the diffusion coefficient of snow.

Let $D_{v-a}$ represent the binary diffusion coefficient of water vapor in air. One view of mass transfer in snow is that water vapor diffusion is driven by the local (microscale) temperature gradient in the humid air constituent. Since the phase transitions that take place at the microscale serve as a temporal storage of vapor in the form of ice, they should, in principle, reduce the effective water vapor transport, and therefore reduce the effective diffusion coefficient. The work of Giddings and LaChapelle (1962), Calonne et al. (2014), Shertzer and Adams (2018), and Fourteau et al. (2021a, 2021b) follow this line of reasoning. In brief, they adopt the view

$$D_s < D_{v-a} \quad .$$

An alternate perspective of mass transfer in snow is that hand-to-hand vapor transport resulting from sublimation and deposition of water vapor is a transport mechanism contributing to the diffusion coefficient, $D_s$. In this context, the ice phase is viewed as a near instantaneous source/sink of water vapor transport, thereby shortening diffusion paths through the humid air and enhancing diffusion rates. The key attribute of this reasoning is that water vapor molecules are indistinguishable from one another. Water vapor condensing on the bottom of an ice grain is identical, in form, to water vapor sublimating off the top of an ice grain. Prior research advocating this position may be found in Yosida (1955), Sommerfeld (1982), Colbeck (1993), and Hansen (2019). This approach suggests that, for low density snow, the diffusion coefficient for snow lies close to the diffusion coefficient for humid air alone with perhaps a slight enhancement under strong temperature gradients (Hansen, 2019).

The paper begins with a comparison of the mathematical framework of the two approaches to the diffusion coefficient outlined above. The comparison is presented in the context of a layered microstructure of ice and humid air, Figure 2. The layered microstructure represents an ideal microstructure to study in that hand-to-hand water vapor transport plays a dominant role in mass transfer. In addition, an analytical solution for the energy flux exists—a solution based only on one-dimensional heat and mass transfer principles with a long history of supporting development.

Next, a rigorous control volume analysis of balance of mass is performed as an independent calculation of the diffusion coefficient. The control volume analysis brings to light three important results: i) the hand-to-hand model of diffusion correctly predicts the diffusion coefficient, ii) volume averaging the local (microscale) mass flux, as presented in Fourteau et al. (2021a), does not capture the



total transport of water moving through the system, and iii) diffusion is enhanced at all ice volume
fractions compared to diffusion through humid air alone.
While the hand-to-hand diffusion analogy is elegant and incredibly valuable in properly modeling
mass diffusion, the fundamental criticism remains that the proposed diffusion mechanism, as put forth by
Yosida (1955), "is not physically sound" (Fourteau et al., 2021a). An entirely fresh perspective on
diffusion is provided where hand-to-hand water vapor transport is dispensed with as a diffusion
mechanism while achieving the same results. In brief, rather than looking at the ice as blocking
microscale diffusion, the ice phase should be viewed as a reservoir of water vapor existing within the
material. Remarkable clarity on mass diffusion in ice/humid air mixtures is achieved in an entirely
different light.
Hand-to-hand vapor transport is also an important mechanism of mass transfer in snow as,
without it, there would be no temperature gradient metamorphism. Hence, the layered microstructure
provides a foundational guide as to how to move forward in studying thermophysical processes in snow.

## 2. Ground truths

In this section, two topics are introduced that provide a valuable foundation for the heat and mass
transfer analysis that follows. The results are noncontroversial and simply represent ground truths
necessary to move forward.
Some basic assumptions are also introduced that are assumed to hold at all times, including:
• Infinitely fast surface kinetics for deposition and sublimation of water vapor are assumed
for the layered microstructure of ice and humid air
• The humid air is saturated
• Convection and radiation are neglected

### 2.1 Defining the mass flux

To begin, a few comments about the nature of flux vectors in general are appropriate. In physics
and applied mathematics, the flux of a vector quantity represents the amount of the vector field passing
through a surface per unit of area per unit of time. Specifically, referring to Figure 3, if $\boldsymbol{n}$ defines a unit
normal for the differential surface, $dS$, and $\boldsymbol{F}$ is a vector field, the flux through the surface is a scalar
given by
$$\text{Flux} = \int_{\partial \mathcal{R}} \boldsymbol{F} \cdot \boldsymbol{n} \; dS \quad , \tag{1}$$

where $\partial \mathcal{R}$ defines the surface. Examples of flux are numerous in mechanics and include phenonena such
as mass flux, momentum flux, and kinetic energy flux. The *flux of mass across the boundary $\partial \mathcal{R}$* is of
interest here, i.e., let $\boldsymbol{F} = \rho \boldsymbol{v}$
$$\text{mass flux} = - \int_{\partial \mathcal{R}} \rho \, \boldsymbol{v} \cdot \boldsymbol{n} \; dS \quad . \tag{2}$$

The minus sign in the above simply indicates mass is leaving the region $\mathcal{R}$.





Now return to Figure 2(a), showing the homogenized layered microstructure in the presence of a temperature gradient, bounded by solid ice blocks held at fixed temperatures. The mass flux across the upper boundary of the layered microstructure is the amount of mass passing through the upper surface per unit of area per unit of time. Physically, it is the amount of water vapor turning to ice at the solid ice/humid air boundary. Note that a humid air layer within the layered microstructure always lies adjacent to the solid ice block.

As time proceeds, ice accretion occurs on the bottom of the bounding upper solid ice block, resulting in an advancing ice front that moves downward with time. Importantly, the appearance of downward motion is entirely the result of upward motion of water vapor and subsequent deposition on the ice surface. Conservation of mass at the solid ice/humid air interface requires

$$\gamma_{\mathrm{v}} v_{\mathrm{v}} = -\gamma_{\mathrm{i}} v_{\mathrm{f}} \quad , \tag{3}$$

where $\gamma_{\mathrm{v}}$ is the water vapor density, $v_{\mathrm{v}}$ is the vapor diffusion velocity, $\gamma_{\mathrm{i}}$ is the density of ice, and $v_{\mathrm{f}}$ is the downward velocity of the accumulating ice front. By tracking the accumulating ice front over time at the upper solid ice boundary, either experimentally or theoretically, one is afforded the remarkable opportunity to quantify the surface mass flux, $\gamma_{\mathrm{v}} v_{\mathrm{v}}$, transcending the upper boundary.

Similarly, the mass flux across the lower boundary is the amount of mass passing through the lower bounding surface per unit of area per unit of time. Physically, it is the amount of ice in the lower solid ice block sublimating to water vapor at the solid ice/humid air boundary. As time proceeds, sublimation off the lower block results in a receding ice front on the lower bounding ice block that moves downward with time. The rate of ice sublimation is also identical to the microscale humid air mass flux, $\gamma_{\mathrm{v}} v_{\mathrm{v}}$.

The conclusion, then, is that the mass transfer moving through the layered ice/humid air system is the same as the mass flux sublimating from the lower solid ice surface and depositing on the upper ice surface and this mass flux is given by $\gamma_{\mathrm{v}} v_{\mathrm{v}}$. Finally, a bit of numerical context is useful here in that the magnitude of $(v_{\mathrm{f}} / v_{\mathrm{v}})$ is on the order of $10^{-6}$.

### 2.2 The energy flux of humid air as a pure substance

The energy flux of humid air as a pure substance follows the classic work on *Transport Phenomena* by Bird et al. (1960). In brief, the total energy flux for humid air may be written as

$$\boldsymbol{q} = \boldsymbol{q}^{(c)} + \boldsymbol{q}^{(d)} \tag{4}$$

where $\boldsymbol{q}^{(c)}$ is the conductive flux and $\boldsymbol{q}^{(d)}$ represents "contribution from the interdiffusion of the various species present." Utilizing Fourier's law for the conductive flux and Fick's law for the diffusive flux (Bird et al., 1960), the 1D energy flux for humid air may be expressed as (Hansen and Foslien, 2015)

$$q = -\left(k_{\mathrm{ha}} + u_{\mathrm{sg}}\left(\frac{d\gamma_{\mathrm{v}}}{d\theta}\right) D_{\mathrm{v-a}}\right)\frac{\partial\theta}{\partial x} \quad , \tag{5}$$

where $k_{\mathrm{ha}}$ is the thermal conductivity, $D_{\mathrm{v-a}}$ is the binary diffusion coefficient of water vapor in air, $u_{\mathrm{sg}}$ is the latent heat of sublimation of ice, and $\theta$ is the temperature. Following Bird et al. (1960) one can identify

$$\text{conductive flux} = -k_{\mathrm{ha}}\frac{\partial\theta}{\partial x} \quad , \tag{6}$$



The Cryosphere



Discussions

and
$$\text{mass flux} = \gamma_v v_v = -D_{v-a}\left(\frac{d\gamma_v}{d\theta}\right)\frac{\partial\theta}{\partial x} \quad . \tag{7}$$
### 3.   Comparing the diffusion coefficient definitions
In order to model the thermophysical processes in a snowpack, knowledge of the macroscale
energy flux for snow is required. The energy flux is given by
$$q_s = -\left(k_s + u_{sg}\left(\frac{d\gamma_v}{d\theta}\right)D_s\right)\frac{\partial\theta}{\partial x} \quad , \tag{8}$$
where $k_s$ is the thermal conductivity and $D_s$ is the diffusion coefficient for snow. While these properties
influence the temperature profile through the snowpack, they also evolve with the changing microstructure
that occurs during snow metamorphism. As such, analytical models for each of these parameters are
sought that can account for microstructural evolution, a lofty goal to be sure.
### 3.1 The layered microstructure
The exact macroscale energy flux density for the layered ice/humid air microstructure is fully
developed in Hansen and Foslien (2015). However, a dramatic simplification of the analytical form of the
energy flux may be achieved by restricting ice volume fractions to be less than 0.8.  This simplified form
of the energy flux is given by
$$q_{lm} = -\left(\left(\frac{k_{ha}}{\phi_{ha}}\right) + \left(\frac{D_{v-a}}{\phi_{ha}}\right)u_{sg}\frac{d\gamma_v}{d\theta}\right)\frac{\partial\theta}{\partial x} \quad , \tag{9}$$
where the subscript "lm" denotes "layered microstructure."  Figure 4 provides a comparison of the exact
energy flux and the approximate energy flux of Eq. (9) at -2°C. The figure shows the exact and
approximate forms of the energy flux are nearly identical for $\phi_i < 0.8$. Furthermore, the approximate
form is most accurate at low densities where diffusion is most prominent. Equation (9) serves as a starting
point for the discussion of the two definitions of the diffusion coefficient.
An important feature of the development of the energy flux of the layered microstructure is that
the energy flux of the macroscale continuum is identical to the energy flux of the ice and humid air
constituents respectively, i.e.,
$$q_{lm} = q_{ha} = q_i \quad . \tag{10}$$
This relationship is used repeatedly to transition from the macroscale to the humid air microscale
### 3.2 The bounding surface flux approach to the diffusion coefficient
The macroscale energy flux of Eq. (9) can be placed into a familiar form for heat transfer in
humid air alone as presented in Section 2.2.  By restricting the ice volume fraction to values below 0.8,
the constituent temperature gradients may be approximated as
$$\left(\frac{\partial\theta}{\partial\xi}\right)_i \approx 0 \qquad \text{and} \qquad \left(\frac{\partial\theta}{\partial\xi}\right)_{ha} = \left(\frac{1}{\phi_{ha}}\right)\frac{\partial\theta}{\partial x} \quad .$$



Noting the above and recognizing the energy flux at the macroscale is identical to the energy flux through
the humid air layer leads to

$$q_{\text{lm}} = q_{\text{ha}} = -\left(k_{\text{ha}} + D_{\text{v-a}}u_{\text{sg}}\frac{d\,\gamma_{\text{v}}}{d\,\theta}\right)\left(\frac{\partial\theta}{\partial\xi}\right)_{\text{ha}}.$$
(11)

Equation (11) is recognized as a precise restatement of Eqs. (4) and (5), defining the energy flux of
humid air as a pure substance following the classic work on *Transport Phenomena* by Bird et al. (1960)—
a fundamental ground truth. Following Bird et al. one can write

$$\text{conductive flux} = -k_{\text{ha}}\left(\frac{\partial\theta}{\partial\xi}\right)_{\text{ha}}$$

$$= -\left(\frac{k_{\text{ha}}}{\phi_{\text{ha}}}\right)\frac{\partial\theta}{\partial x}\ ,$$
(12)

and

$$\text{mass flux} = -D_{\text{v-a}}\left(\frac{d\gamma_{\text{v}}}{d\theta}\right)\left(\frac{\partial\theta}{\partial\xi}\right)_{\text{ha}}$$

$$= -\left(\frac{D_{\text{v-a}}}{\phi_{\text{ha}}}\right)\left(\frac{d\gamma_{\text{v}}}{d\theta}\right)\frac{\partial\theta}{\partial x}\ .$$
(13)

Note that the conductive flux and the mass flux identified above are correct for the macroscale
layered continuum as well as the microscale of the pure humid air layer. Specifically, the mass flux of Eq.
(13) is identical to the surface flux of water vapor crossing the boundaries at the interface of the solid
ice/humid air mixture—at the upper boundary in the form of deposition and ice accretion as well as at the
lower boundary in the form of sublimation. In other words, Eq. (13) represents the mass flux moving
through the ice/humid air mixture.
Consistent with the above discussion, the conductive heat flux and mass flux lead to following
definitions of thermal conductivity and the diffusion coefficient given by

$$k_{\text{lm}} = \left(\frac{k_{\text{ha}}}{\phi_{\text{ha}}}\right)\ .$$
(14)

and

$$D_{\text{lm}} = \left(\frac{D_{\text{v-a}}}{\phi_{\text{ha}}}\right)\ .$$
(15)

While the specific forms of the conductive flux and mass flux given in Eqs. (12) and (13) may
seem intuitively obvious for a layered ice/humid air microstructure and further represent a ground truth
for energy transfer in humid air as a pure substance, they are at the very heart of the historical (and
current) controversy surrounding the diffusion coefficient.
Forteau et al. (2021a) argue that the decomposition of Eq. (9) into a conductive flux and a mass
flux defined by Eqs. (12) and (13) is not unique and other decompositions exist. In particular, their
arguments focus on volume averaging the local (microscale) mass flux to obtain the macroscale mass
flux.





### 284   3.3 A volume averaged approach to the diffusion coefficient
Fourteau et al. (2021a) present arguments that the diffusion coefficient may be computed by
volume averaging the diffusion through the humid air phase and assuming the ice volume does not
contribute to diffusion. In the context of the layered microstructure, volume averaging the local mass flux
of Eq. (13) over the entire volume leads to a macroscale diffusion coefficient given by
$$D_{\mathrm{lm}} = D_{\mathrm{v-a}} \quad . \tag{16}$$
Noting the energy flux of Eq. (9), the above definition of the diffusion coefficient leads to
a definition of thermal conductivity given by (see Eq. C7, Appendix C, Fourteau et al., 2021a)
$$k_{\mathrm{lm}} = \left( \frac{\left( k_{\mathrm{ha}} + \phi_i D_{\mathrm{v-a}} \ u_{\mathrm{sg}} \left( \frac{d\gamma_v}{d\theta} \right) \right)}{\phi_{\mathrm{ha}}} \right) \quad . \tag{17}$$
The above thermal conductivity and diffusion coefficient decomposition suggested by Fourteau et
al. (2021a, 2021b), while a correct mathematical decomposition of the energy flux, has some troubling
aspects related to the physics of heat and mass transfer in the layered microstructure including:
• The diffusion coefficient of Eq. (16) does not predict the known mass transport of water vapor
leaving the upper boundary of Figure 2(a) in the form of ice accretion on the upper solid ice
block, or water vapor crossing the lower boundary in the form of sublimation from the lower
ice block. Moreover, it clearly does not represent the total mass transfer due to diffusion as the
thermal conductivity of Eq. (17) also contains a diffusion term.
• The thermal conductivity of Eq. (17) does not represent a true thermal conductivity for the
layered microstructure, which is correctly defined by Eq. (14). Equation (14) is simply a
statement of ground truth for thermal conductivity of humid air as a pure substance as outlined
in Section 2.2. In brief, why should the thermal conductivity of humid air as a pure substance
involve diffusion.
The lack of physical meaning of the conductive flux and the mass flux of the approach of
Fourteau et al. (2021a, b) may be traced to their definition of the diffusion coefficient based on a volume
average of the local (microscale) diffusion velocity. A precise explanation as to why the volume
averaging of the mass flux put forth by Fourteau (2021a) fails to model the mass flux across the
boundaries is provided in Section 4.3.
### 317   3.4 The role of hand-to-hand water vapor transport in the macroscale heat and mass
### 318       transport properties
Sections 3.2 and 3.3 lay out two separate views of mass diffusion occurring in a layered
ice/humid air microstructure. In what follows, additional physical insight into the connection and
fundamental differences of the two approaches is provided. In doing so, the role of hand-to-hand water
vapor transport in mass diffusion is revealed.
Again, begin with the normalized energy flux of the layered microstructure taken from Eq. (9)
and written as





$$\frac{q}{\left(\frac{\partial \theta}{\partial x}\right)} = -\left(\frac{k_{\text{ha}}}{\phi_{\text{ha}}} + \left(\frac{D_{\text{v}-\text{a}}}{\phi_{\text{ha}}}\right) u_{\text{sg}} \left(\frac{d\,\gamma_{\text{v}}}{d\,\theta}\right)\right).$$
(18)

The second term on the RHS of the above involving diffusion may be broken out into two terms
weighted by volume fractions of ice and humid air leading to
$$\frac{q}{\left(\frac{\partial \theta}{\partial x}\right)} = -\left(\underbrace{\frac{k_{\text{ha}}}{\phi_{\text{ha}}}}_{(1)} + \underbrace{\phi_{\text{i}}\left(\frac{D_{\text{v}-\text{a}}}{\phi_{\text{ha}}}\right) u_{\text{sg}}\left(\frac{d\,\gamma_{\text{v}}}{d\,\theta}\right)}_{(2)} + \underbrace{\phi_{\text{ha}}\left(\frac{D_{\text{v}-\text{a}}}{\phi_{\text{ha}}}\right) u_{\text{sg}}\left(\frac{d\,\gamma_{\text{v}}}{d\,\theta}\right)}_{(3)}\right).$$
(19)


The approach of Fourteau et al. (2021a) presented in Section 3.3 combines Terms 1 and 2 of Eq.
(19) to arrive at thermal conductivity and diffusion coefficient definitions given by
$$k_{\text{lm}} = \left(\frac{k_{\text{ha}}}{\phi_{\text{ha}}}\right) + \phi_{\text{i}}\left(\frac{D_{\text{v}-\text{a}}}{\phi_{\text{ha}}}\right) u_{\text{sg}}\left(\frac{d\,\gamma_{\text{v}}}{d\,\theta}\right),$$
(20)

and
$$D_{\text{lm}} = D_{\text{v}-\text{a}}.$$
(21)

As a general observation, Terms 2 and 3 of Eq. (19) clearly involve mass transfer involving the
diffusion coefficient of water vapor in air. As noted previously, logic would suggest that these terms be
grouped together, rather than combining a mass diffusion term with thermal conductivity as done in Eq.
(20).

In contrast, the approach of grouping the similar diffusion terms, Term 2 & 3 of Eq. (19), is
followed in Section 3.2, leading to the thermal conductivity and diffusion coefficient having the
definitions of
$$k_{\text{lm}} = \left(\frac{k_{\text{ha}}}{\phi_{\text{ha}}}\right),$$
(22)

and
$$D_{\text{lm}} = \left(\frac{D_{\text{v}-\text{a}}}{\phi_{\text{ha}}}\right).$$
(23)

The above definitions are developed from a macroscale energy flux of Eq. (9) that is identical to
the energy flux of humid air as a pure substance at the microscale. Equation (23), and its associated mass
flux given by Eq. (13) also represents the true mass transfer across the upper and lower boundaries of
Figure 2(a). Finally, note the striking similarities in Eqs. (22) and (23) and their elegant simplicity.
The differences in the approaches of Sections 3.2 and 3.3 clearly fall to the second term of Eq.
(19). The fundamental question, then, is "what is the precise physical significance of the second term?"
The answer is that this term is the mass diffusion and heat transfer associated with hand-to-hand analogy
of water vapor transport involving the simultaneous condensation and sublimation of water vapor in the
ice phase. To show this physically, note that the volume fractions of ice and humid air are identical to the



lineal fraction for a test line of length, $L$, passing through the microstructure. Hence Terms 2 and 3 in Eq. (19) may be combined to show the heat flux due to mass diffusion is

$$\text{diffusion heat flux} = -\left[\frac{L_i}{L}\left(\frac{D_{v-a}}{\phi_{ha}}\right)u_{sg}\left(\frac{d\,\gamma_v}{d\,\theta}\right) + \frac{L_{ha}}{L}\left(\frac{D_{v-a}}{\phi_{ha}}\right)u_{sg}\left(\frac{d\,\gamma_v}{d\,\theta}\right)\right]\frac{\partial\theta}{\partial x}, \qquad (24)$$

where $L_i$ and $L_{ha}$ are the respective lengths of a test line passing through the ice phase and the humid air phase. The associated mass flux is given by

$$\text{mass flux} = -\left(\underbrace{\frac{L_i}{L}\left(\frac{D_{v-a}}{\phi_{ha}}\right)}_{(A)} + \underbrace{\frac{L_{ha}}{L}\left(\frac{D_{v-a}}{\phi_{ha}}\right)}_{(B)}\right)\left(\frac{d\,\gamma_v}{d\,\theta}\right)\frac{\partial\theta}{\partial x} \qquad (25)$$

Term B in Eq. (25) represents the mass flux due to water vapor diffusion through the humid air scaled by the normalized length of a humid air test line. It is this term that Fourteau et al. (2021a) have identified as the diffusive mass flux for the macroscale layered microstructure.

Term A in Eq. (25) may be viewed as the mass flux from hand-to-hand water vapor transport by the ice phase as a result of continuous condensation and sublimation. Physically, with regard to hand-to-hand water vapor transport, the ice phase can only transfer mass as fast as it receives it from the humid air and this is precisely governed by the humid air mass flux at the microscale given by

$$\gamma_v v_v = \left(\frac{D_{v-a}}{\phi_{ha}}\right)\left(\frac{d\,\gamma_v}{d\,\theta}\right)\frac{\partial\theta}{\partial x}. \qquad (26)$$

The above result is then scaled by the ice lineal ice fraction ($L_i/L$) for the layered microstructure to account for the distance covered by the ice phase as vapor hop scotches across the ice phase. As soon as vapor arrives at a lower ice surface, an equivalent amount leaves the upper surface. The end result is precisely Term A in Eq. (25).

While the above discussion provides a cogent physical explanation of the role of hand-to-hand vapor transport in the diffusion coefficient, one may argue that the discussion lacks the necessary mathematical rigor to be wholly defensible. This weakness is dispelled in Sections 4 & 5 through a rigorous control volume analysis, as well as tracking a material point of water throughout its life in traveling through the microstructure to the upper solid ice boundary.

4.  **Control volume analysis**

The continuum forms of the governing balance equations of mass, momentum, and energy are developed in terms of a material volume—a region in space containing a known quantity of mass. As the body is deformed, the material volume moves in space and may also deform in shape. Moreover, in the case of fluids, the configuration of the deformed body is generally not known until the problem is solved.

To alleviate the challenges of studying a material volume, the idea of a control volume is introduced where one defines a region in space to apply the governing equations. Sonin (2003) provides an excellent discussion of control volumes stating: "*A control volume is an arbitrarily defined volume with a closed bounding surface (the control surface) that separates the universe into two parts: the part contained within the control volume, and the rest of the universe. The control surface is a mental construct, transparent to all material motion, and may be static in the chosen reference frame, or moving*





*and expanding or contracting in any specified manner. The analyst specifies the velocity v(r,t) at all*
*points of the control surface for all time."* The selection of a control volume is driven by the information
that is desired.

In this section, two control volume analyses are presented for the layered ice/humid air
microstructure, including a fixed control volume and a moving control volume that moves downward in
lockstep with the downward advancing ice front on the top boundary, see Figure 5. For brevity, in the
following discussion, the upper and lower ice blocks are referred to as "solid ice" while the layered
ice/humid air microstructure is simply referred to as the "ice mixture."

Let the fixed control volume and the moving control volume be coincident at time $t = 0$ as
shown in Figure 5(a). At a later time, the moving control volume has diverged from the fixed control
volume as it tracks the moving ice front formed by ice accretion at the upper boundary, Figure 5(b). Note
that, as $t \to \infty$, the ice phase would advance sufficiently such that the entire fixed control volume would
enclose ice only.

**4.1 Mass flux of humid air as a pure substance**

The mass flux of humid air as a pure substance provides an important foundation for
understanding the mass flux of the layered microstructure. Figure 6(a) shows the advancing ice front from
the upper boundary due to ice accretion from water vapor transport due to diffusion in humid air alone.
Now introduce a characteristic time, $\tau$, at which the advancing ice front at the upper boundary has moved
a length, $\ell$. The total mass contained in the advancing ice front may be expressed in several forms given
by:

$\gamma_v \hat{v}_v \tau = -\gamma_i \hat{v}_f \tau = \gamma_i \ell$ ,                     (27)

where the hats above the velocity symbols are used to refer to humid air as a pure substance. The
characteristic time and length $(\tau, \ell)$ for humid air as a pure substance serve as a valuable baseline for the
analysis of the layered microstructure.

A subtle but important observation throughout the analysis of this section is that the true water
vapor diffusion velocity is unaffected by the speed of the advancing ice front as $(\hat{v}_f / \hat{v}_v)$ is on the order
of $10^{-6}$.

Fluid mechanics is replete with solutions involving moving control volumes and these moving
control volumes often track the motion of a moving front. In the present case, the moving control volume
tracks the advancing ice front at the upper boundary and the receding ice front at the lower boundary, (the
green control volume of Figure 5(b)). The following fundamental properties of the analysis are observed
for humid air as a pure substance:

1) The mass of the control volume is constant in time implying

$\frac{d}{dt} \int_{\mathcal{R}} \rho \, dV = 0$ .               (28)

The Reynold's Transport Theorem for mass conservation may be expressed in the form
(Sonin, 2003)

$\frac{d}{dt} \int_{\mathcal{R}(t)} \rho \, dV + \int_{\partial \mathcal{R}(t)} \rho (\hat{v}_v - v_C) \, dV = 0$ ,       (29)





where $v_C$ is the control surface velocity. Recognizing $v_C = v_f$, the transport theorem for water vapor reduces to

$$\int_{\partial \mathcal{R}(t)} \gamma_v (\hat{v}_v - v_f)\, dV = 0 \quad . \tag{30}$$

Noting $(v_f / \hat{v}_v)$ is on the order of $10^{-6}$, there follows

$$\int_{\partial \mathcal{R}(t)} \gamma_v \hat{v}_v\, dV = 0 \quad . \tag{31}$$

The above simply implies the mass of water vapor entering the control volume from below is equal to the mass of water vapor leaving the control volume from above.

2) The control volume boundaries continuously lie at the interface between the solid ice and the ice mixture. Hence, mass transfer across the control volume surface is precisely the mass flux of water vapor crossing the boundaries between the humid air and the bounding solid ice. The mass flux across the upper and lower boundaries is governed by

$$\text{mass flux} = \gamma_v \hat{v}_v = D_{v-a} \left( \frac{d\gamma_v}{d\theta} \right) \frac{\partial \theta}{\partial x} \quad . \tag{32}$$

Sublimation is occurring at the lower boundary while deposition of water vapor is occurring at the upper boundary.

**4.2 Layered microstructure: moving control volume**

As in the case of humid air as a pure substance, let the moving control volume track the moving ice front at the boundaries. The following fundamental properties of the analysis are observed:

1) The mass of the control volume is constant in time implying yielding Eq. (28). Following identical arguments to those for humid air as a pure substance, the mass flux across the control surface may be written as

$$\int_{\partial \mathcal{R}(t)} \gamma_v v_v\, dV = 0 \quad , \tag{33}$$

implying the mass of water vapor entering the control volume from below is equal to the mass of water vapor leaving the control volume from above.

2) The control volume boundaries continuously lie at the interface between the solid ice and the ice mixture. Hence, mass transfer across the control volume surface is precisely the mass flux of water vapor transcending the boundaries between the ice mixture and the bounding solid ice. The mass flux across the upper and lower boundaries is enhanced due to the elevated humid air temperature gradient and is governed by Eq. (13) as

$$\text{mass flux} = \gamma_v v_v = \left( \frac{D_{v-a}}{\phi_{ha}} \right) \left( \frac{d\gamma_v}{d\theta} \right) \frac{\partial \theta}{\partial x} \quad . \tag{34}$$

Comparing Eqs. (32 & 34) one can write

$$\gamma_v v_v = \frac{\gamma_v \hat{v}_v}{\phi_{ha}} \quad . \tag{35}$$



A comparison of the mass flux for the layered microstructure versus the mass flux of humid air
alone is most readily followed through a choice of specific constituent volume fractions. Hence, consider
an ice volume fraction of $\phi_i = 1/3$, implying $\phi_{ha} = 2/3$. For this case, the mass flux of the layered
microstructure given in Eq. (35) is 1.5 times the mass flux of humid air as a pure substance. Furthermore,
for the characteristic time, $\tau$, the ice front advancing from the upper boundary has moved $1.5\ell$ compared
to $\ell$ for humid air as a pure substance, Figure 6(b).
Now consider a unit cell for a *moving control volume* advancing with the ice front shown in
Figure 7(a). Further, define a local coordinate system ($\xi$) moving downward with the control volume.
Hence, the coordinate system is moving downward at the speed of the accumulating ice front on the lower
boundary of the unit cell. Two interesting observations fall out:
• The problem is steady state relative to the moving reference frame meaning the configuration of
the unit cell is unchanged with time. Hence, the mass within the control volume is time
independent, as is the surface flux across the boundaries of the unit cell.
• Relative to an observer on the control volume, *an arbitrary material point in the ice phase is*
*seen moving upwards toward the upper surface of the ice* with a velocity of $v_{i/C} = -v_f$, where
$v_{i/C}$ is the velocity of material point with respect to the control volume.
A mass balance at the solid vapor interface in the unit cell yields
$\gamma_i v_{i/C} = -\gamma_i v_f = \gamma_v v_v$ .                                          (36)
The mass flux across the upper and lower boundaries of the unit cell is given by
mass flux $= \gamma_v v_v = \left(\dfrac{D_{v-a}}{\phi_{ha}}\right)\left(\dfrac{d\gamma_v}{d\theta}\right)\dfrac{\partial\theta}{\partial x}$ .                (37)
The volume average of the mass flux over the entire volume of the unit cell is given by
mass flux $= \phi_{ha}\,\gamma_v\,v_v + \phi_i\gamma_i v_{i/C}$
$= (\phi_{ha} + \phi_i)\,\gamma_v\,v_v = \gamma_v\,v_v = \left(\dfrac{D_{v-a}}{\phi_{ha}}\right)\left(\dfrac{d\gamma_v}{d\theta}\right)\dfrac{\partial\theta}{\partial x}$ .                (38)
Importantly, the volume averaged mass flux and the surface mass flux agree. Furthermore, the
term $\phi_i\gamma_i v_{i/C} = \phi_i\gamma_v v_v$ in Eq. (38) is numerically equal to the hop scotching effect of hand-to-hand
vapor transport described in Section 3.4.
An additional appealing aspect of this moving control volume analysis is that the configuration of
the unit cell does not matter. For instance, consider the unit cell of Figure 7(b) where the unit cell
boundaries extend through the ice phase. Equations (36-38) remain valid and the configuration of the unit
cell remains steady-state (time independent).
***Perhaps the most important aspect of the unit cell analysis of the moving control volume is that***
***the ice phase is a contributing factor to the overall mass transport of water moving through the system.***
As a result, diffusion of water vapor is enhanced at all humid air volume fractions. This result is at odds



with the authors who suggest the only role of the ice phase is to occupy volume where diffusion cannot
occur, an untenable position regarding mass transfer in the layered microstructure.
**4.3 Layered microstructure: fixed control volume**
Now consider the mass transfer analysis of the layered microstructure using a fixed control
volume shown by the red dashed line in Figure 5(b)—a decidedly more complicated approach. Note that
the advancing ice front moves downward into the control volume due to ice accretion on the solid ice/ice
mixture boundary. The mass flux at the upper solid ice/ice mixture boundary is governed by Eq. (34).
Before discussing mass transfer through the lower boundary of the control volume, let us examine
a unit cell in the context of a fixed control volume. Figure 8(a) shows a unit cell at time $t = 0$. As time
proceeds, the ice phase advances downward due to condensation and sublimation on the upper and lower
ice boundaries, respectively. Figure 8(b) shows this downward advancing ice mass at a later time.
Eventually the ice mass will pass through the lower boundary as it simultaneously reappears at the upper
boundary.
Three important observations governing mass transfer in the fixed control volume unit cell are:
• While the ice phase *appears* to be propagating downward, it is caused by water vapor
moving upward with a monotonically increasing (nonnegative) displacement. *All* material
points of water in the system are either diffusing upward through the humid air or at zero
velocity while locked in the ice phase. Water material points never have a negative
velocity, despite the ice phase *appearance* of downward motion.
• The volume average of the humid air mass flux within the unit cell is given by

$$\langle \gamma_v \, v_v \rangle = D_{v-a} \left( \frac{d\gamma_v}{d\theta} \right) \frac{\partial \theta}{\partial x} . \tag{39}$$

• The surface flux across control volume boundaries is non-steady but periodic. In other
words, at times the ice phase will block vapor transport across a control volume
boundary while at other times vapor will pass through a humid air boundary of the
control volume. A temporal average over one period of the surface flux over either the
upper or lower boundaries reveals a flux identical to the volume average given by

$$\overline{\gamma_v \, v_v} = D_{v-a} \left( \frac{d\gamma_v}{d\theta} \right) \frac{\partial \theta}{\partial x} . \tag{40}$$

Hence, the time averaged surface flux over the upper and lower boundaries of the unit cell agrees with
volume averaged mass flux.
One is now faced with an interesting paradox that strikes at the heart of the debate over the
definition of the diffusion coefficient. One can summarize the conflict with the following observations
made for the fixed control volume of Figure 5(b):
• The rate of mass transfer in the form of ice accretion at the solid ice/ice mixture upper
boundary is given by Eq. (34) as:

$$\gamma_v \, v_v = \frac{\gamma_v \hat{v}_v}{\phi_{ha}} = \frac{D_{v-a}}{\phi_{ha}} \left( \frac{d\gamma_v}{d\theta} \right) \frac{\partial \theta}{\partial x}$$




- The volume average microscale mass flux is given by Eq. (39) as

$$\langle \gamma_v\, v_v \rangle = \gamma_v \hat{v}_v = D_{v-a}\left(\frac{d\gamma_v}{d\theta}\right)\frac{\partial \theta}{\partial x}\,.$$

- The temporal average flux across the lower boundary of the fixed control volume is given by Eq. (40), i.e.

$$\overline{\gamma_v\, v_v} = \gamma_v \hat{v}_v = D_{v-a}\left(\frac{d\gamma_v}{d\theta}\right)\frac{\partial \theta}{\partial x}$$

*An apparent conflict arises in that the mass flux crossing the upper solid ice/ice mixture boundary exceeds the mass flux entering the control volume from the lower surface which is also equal to the volume average of the microscale mass flux.*

The conflict is resolved through a careful examination of the role of the ice phase that exists in the form of layering within the fixed control volume. Specifically, individual layers of ice act as large reservoirs that release water vapor as needed, allowing the water vapor to diffuse through the humid air.

Consider a fixed control volume as shown in Figure 9(a). The reservoirs of water vapor contained in the ice layers in the control volume disappear over time as they effectively restore the mass imbalance across the control volume boundaries described above. For example, suppose at time $t = 0$, there are 50 layers of ice within the fixed control volume of Figure 9(a). At a later time, as the ice front advances downward from the solid ice upper boundary, there may only be 40 layers. At still a later time, 30 layers will exist and so on. Eventually, all layers of ice will have vanished through diffusion in the humid air, arriving at the upper solid ice boundary in the form of ice accretion. In brief, the ice phase is definitely contributing to the mass transfer through the layered microstructure. In fact, the ice phase is a major source of water vapor while the humid air acts as the transport mechanism.

The opposite phenomenon is occurring in the fixed control volume of Figure 9(b). Here, sublimation at the lower solid ice/ice mixture boundary has a mass flux entering the layered microstructure at the rate of Eq. (34). The mass flux leaving the upper boundary is defined by the surface flux of Eq. (40), also equal to the humid air mass flux volume average of Eq. (39). *An apparent conflict again arises in that the mass flux crossing the lower solid ice/ice mixture boundary exceeds the mass flux leaving the control volume from the upper surface of Figure 9(b).*

The conflict is again resolved by the presence of the ice layering. Whereas, in the control volume of Figure 9(a) where ice layers disappear over time, in the control volume of Figure 9(b), ice layers grow in number over time. For example, suppose at time $t = 0$, there are 50 layers of ice within the fixed control volume of Figure 9(b). At a later time, as the solid ice recedes at the lower boundary, the number of layers may rise to 60 layers. At still a later time, 70 layers will exist and so on. Note that the total number of layers in the entire system, defined as the sum of layers in Figure 9(a) and 9(b) is constant as the problem is steady state.

Just like the analysis of the moving control volume, in the case of the fixed control volume analysis, ***the ice phase is a major contributing factor to the overall mass transport of water moving through the system.*** This "reservoir phenomenon" explains why the simple volume averaging of the humid air microscale mass flux proposed by Fourteau et al. (2021) does not correctly capture the mass flux of water "moving" through the system.



As a numerical example of the ice reservoir effect, consider an ice volume fraction of $\phi_i = 1/3$,
implying $\phi_{ha} = 2/3$. Further, assume the control volume has length 1.5 $L$ such that an advancing ice
front will fill the control volume in time $\tau$, Figure 9(a). At time $t = 0$, the ice phase present in the mixture
in the form of layering has a total mass of
$$\phi_i \gamma_i 1.5\ell = 0.5\gamma_i\ell \quad . \tag{41}$$
As noted previously, the appearance of individual ice layers moving in a downward direction
causing a counterflux of mass is a mirage, as water (ice or vapor) is always moving upward in a
monotonically increasing (nonnegative) fashion. In particular, water vapor is either advancing toward the
upper boundary in the form of diffusion through the humid air or stationary as solid ice, waiting to reach
the surface of a layer to take off again. The implication of this observation is that all mass within the ice
layers of the control volume at $t = 0$ will reach and become part of the advancing ice front at the solid
ice/ice mixture boundary.
At the same time that the ice layers within the control volume are contributing to the mass flux
over time $\tau$, additional mass enters the fixed control volume from below according to Eq. (39) at the time
averaged rate of $\gamma_v \hat{v}_v$, identical to the mass flux of humid air as pure substance. From Eq. (27) of Section
4.1, the mass added to the fixed control volume by crossing the lower boundary in time $\tau$ is
$$\gamma_v \hat{v}_v \tau = -\gamma_i \hat{v}_f \tau = \gamma_i\ell \tag{42}$$
The total mass of the advancing front at the solid ice/ice mixture boundary is the sum of Eqs. (41) and
(42) given by $\gamma_i 1.5\ell$, making the control volume solid of Figure 9(a) solid ice.
In brief, mass added across the fixed control volume lower surface plus additional ice mass
present in the control volume in the form of ice reservoirs equals the total mass of ice of the advancing ice
front. *Or, in terms of diffusion, the mass flux attributed to the layered ice within the fixed control volume*
*plus the mass flux crossing the lower boundary of the fixed control volume equals the total mass flux*
*across the upper solid ice/ice mixture boundary.* In brief, the layered ice within the control volume should
be viewed as a reservoir of water vapor that enhances diffusion rather than a temporal storage of water
vapor slowing diffusion.
The complexities of the fixed control volume are subtle and require attention to detail. Of course,
all of these complexities can be dispensed with by formulating the mass transfer problem in terms of a
moving control volume as was done in Section 4.2. In either the case of the moving control volume or the
fixed control volume, the mass transfer across the solid ice/ice mixture boundary is identical—physics
demands a solution that is independent of the control volume selected. Furthermore, water vapor diffusion
is enhanced at all ice volume fractions compared to diffusion through humid air as a pure substance. The
diffusion enhancement is identical to the results predicted by the hand-to-hand diffusion analogy put forth
by Yosida (1955).
**5.   The ice phase as a reservoir of water vapor**
A fresh look at the diffusion problem allows one to dispense with hand-to-hand water vapor
transport as a diffusion mechanism. Figure 10 demonstrates the motion of two water vapor material
points, *A* and *B,* over a period of time sufficient for an ice layer to completely turn over its entire mass.
The linear upward sloping portion of the water vapor displacement of *A* and *B* represents time traveling
through humid air whereas the long constant period (zero velocity) represents time residing in the ice
phase. The water vapor transport cycle through a unit cell is complete at $t = t'$ when the water material



points are located at $A'$ and $B'$ and the cycle then repeats. Note that the concept of a counterflux of mass is
a myth in that the motion of water material points $A$ and $B$ is a monotonically increasing function for all
time.
Figure 10 also shows that as point $A$ arrives at the bottom of the ice at $A'$, point $B$ is ready to take
off through humid air at $B'$, just as occurred at $t = 0$. This phenomenon is precisely a description of hand-
to-hand vapor transport.
An alternative view of hand-to-hand diffusion is presented through a careful extrapolation of the
displacement/time history of Figure 10. If one tracks a single material point, $A$, it is clear that all vertical
motion of the point occurs as water vapor diffusing upward through humid air. This observation is, in
some sense, consistent with Fourteau et al. (2021a) suggesting diffusion is controlled by the microscale
diffusion within humid air. Furthermore, the notion that hand-to-hand water vapor transport is
nonphysical in the context of diffusion is removed, i.e., the path of point $A$—in moving from its location
at $t = 0$ to the upper boundary—never involves hand-to-hand water vapor transport.
The reservoir phenomenon is on brilliant display when one examines a moving control volume
described in Section 4.2. In this case, a beautiful analogy of the role of the ice is that of a lake with a
single inlet at one end and a single outlet at the other. Under steady state conditions, the inflow and
outflow to the lake have identical mass flow rates. If one adopts the hand-to-hand model of mass
transport, the lake acts as an instantaneous source/sink for the mass flow rate, just as the ice does in the
layered microstructure.
One can also avoid the hand-to-hand concept of mass transfer in the lake by recognizing the
effective 1D mass flow rate through the lake is identical to the inlet and outlet mass flow rates. While the
velocity within the lake is extremely low, the massive volume of water moving, albeit extremely slowly,
produces the same mass flow rate. In the case of mass transfer through the layered microstructure the
*velocity of ice with respect to the moving control volume* is an identical effect.
The fundamental difference in the two approaches to diffusion described in this paper then, is
that, rather than looking at the ice as blocking microscale diffusion, the ice phase should be viewed as an
existing reservoir of water vapor. If one returns to the path of material point $A$, the extended time spent in
the ice should not be seen as slowing diffusion, rather, point $A$ resides in the reservoir of ice until needed,
when it reaches the upper surface through sublimation of the ice above it. Once point A reaches the upper
surface of a layer, it then sublimates and moves upward through classic diffusion in humid air until it
reaches the next layer (reservoir). Also, because of elevated temperature gradients in the humid air
layered microstructure, water vapor released from the ice travels through the humid air at an enhanced
diffusion velocity compared to the velocity through humid air as a pure substance.
**5.1 A specific example of the reservoir effect**
Let us briefly address the physical arguments put forth in Section 2 of Forteau et al. (2021a)
regarding diffusion in a layered microstructure. To begin, we focus on the red and orange molecules of
Figure 1 of Fourteau et al. (2021a). In reference to the hand-to-hand mass transfer analogy they note:
"*For this mechanism to explain the experimental observations, the continuous deposition and sublimation
should produce a real mass flux from one can to the other, as if the depositing molecule reappeared as
the sublimating one. However, what actually happens is that the depositing molecule (represented as an
orange dot in Fig. 1) remains incorporated at the bottom of the ice grain, thus remaining in the first can.*"
This statement is not true for, if one tracks the motion of the orange molecule over time, the water vapor
molecule remains stationary within the ice until it reaches the surface through sublimation of the ice
above it. At that point, the orange vapor molecule is released via sublimation and allowed to again



transfer upward, de facto moving from the lower can to the upper can—the classic form of diffusion of
water vapor.

In what follows, we present a specific calculation to show that the path of a material point of
water through the layered microstructure of Figure 11 results in enhanced diffusion compared to diffusion
through humid air alone—a nonintuitive result.
Consider the diffusion life of the orange material point shown at point $F$ in Figure 11 and the time
history taken to reach the upper solid ice/ice mixture interface. Let the unit cell have a length dimension
of one as a matter of convenience as the volume fractions then correlate to lengths, i.e.,
$$L_\alpha = \phi_\alpha \; .$$  (43)

The total distance point F must move to reach the upper surface is given by $(2\,L_{\mathrm{ha}})$. Note that the
distance through the ice phase does not enter this calculation because, as the ice phase of the layer
sublimates away, ice is also condensing on the solid ice/ice mixture interface at the same rate.
The total time for the water material point at $F$ to reach the upper surface of the solid ice/ice
mixture is the time required to traverse through the humid air plus the time while at rest and locked in the
ice phase. The humid air diffusion time is given by
$$t_{\mathrm{ha}} = \frac{(2\,L_{\mathrm{ha}})}{v_{\mathrm{v}}} \quad .$$  (44)

Recall that the diffusion velocity is elevated due to the elevated temperature gradient in the humid air of
the layered microstructure and may be computed using Eq. (26).
To compute the total time the material point $F$ resides in the ice phase, one must compute the
time it takes for the sublimating ice at the top of the layer to reach point $F$, currently residing at the
bottom of the ice layer.  From Eq. (36), conservation of mass at the upper ice/humid air interface of the
ice layer leads to
$$-\gamma_{\mathrm{i}}\,v_{\mathrm{f}} = \gamma_{\mathrm{v}} v_{\mathrm{v}} \quad ,$$  (45)

where $v_{\mathrm{f}}$ is the velocity of the receding front of the upper surface of the ice layer. The total time that the
point $F$ resides locked in the ice phase is given by
$$t_{\mathrm{i}} = \frac{-L_{\mathrm{i}}}{v_{\mathrm{f}}} = \frac{(L_{\mathrm{i}})}{v_{\mathrm{v}}}\left(\frac{\gamma_{\mathrm{i}}}{\gamma_{\mathrm{v}}}\right)$$  (46)

The total time for the material point at $F$ to reach the upper solid ice/ice mixture surface is then

$$\tau = t_{\mathrm{ha}} + t_{\mathrm{i}}$$

$$= \frac{(2\,L_{\mathrm{ha}})}{v_{\mathrm{v}}} + \frac{(L_{\mathrm{i}})}{v_{\mathrm{v}}}\left(\frac{\gamma_{\mathrm{i}}}{\gamma_{\mathrm{v}}}\right)$$  (47)

When point $F$ reaches the solid ice/ice mixture interface, thereby ending its travels, the amount of
mass per unit area reaching the upper surface in the form of deposited ice is given by
$$m = \gamma_{\mathrm{v}}(2\,L_{\mathrm{ha}}) + \gamma_{\mathrm{i}} L_{\mathrm{i}} \quad .$$  (48)






Now consider humid air only under the same macroscale temperature gradient as the layered
microstructure, Figure 6(a). One can compare the mass transfer rates between humid air alone and the
layered microstructure in two different ways: i) compute the time required to achieve the same transfer of
mass, or ii) fix the time and compute the quantity of mass that reaches the solid ice/ice mixture boundary
in the form of deposition.
Let us begin by fixing the mass according to Eq. (48) and compute the time required to achieve
this mass transfer for humid air alone. To begin, the length, $d$, of a column of humid air needed to achieve
the total mass of Eq. (48) is given by
$$d = \frac{m}{\gamma_v} = (2\,L_{ha}) + \left(\frac{\gamma_i}{\gamma_v}\right)L_i \qquad . \tag{49}$$
As noted previously, the diffusion velocity is elevated in the humid air/ice mixture compared to
the humid air alone due to the elevated temperature gradients in the layered ice mixture. The diffusion
velocities are related as
$$\hat{v}_v = v_v\,\phi_{ha} \qquad , \tag{50}$$
where the "hat" is used to reference the humid air alone.
The time required to accumulate the mass of Eq. (49) on the bounding upper ice surface is given
by
$$\hat{\tau} = \frac{d}{\hat{v}_v} = \frac{\left((2\,L_{ha}) + \left(\frac{\gamma_i}{\gamma_v}\right)L_i\right),}{\hat{v}_v} \qquad , \tag{51}$$
or noting Eqs. (47 & 50),
$$\hat{\tau} = \frac{\tau}{\phi_{ha}}. \tag{52}$$
The above shows that diffusion in the layered microstructure is enhanced at all humid air volume
fractions as it takes a longer time to achieve the same mass transfer in humid air alone.
If on the other hand, one fixes the time, $\tau$, according to Eq. (47), the total mass crossing the
boundary in the humid air alone is given by
$$\hat{m} = \tau\gamma_v\hat{v}_v = \left(\frac{\left(2L_{ha} + L_i\left(\frac{\gamma_i}{\gamma_v}\right)\right)}{v_v}\right)\gamma_v\,\hat{v}_v$$
$$= \phi_{ha}\,m. \; . \tag{53}$$
Hence, for a fixed time, $\tau$, the mass transfer moving through the system of humid air alone is reduced
compared to the mass transfer in the layered ice/humid air mixture.
Using either approach above, diffusion is enhanced in the layered microstructure and the
diffusion coefficient of the layered microstructure may be expressed precisely as
Eq. (23), i.e.,





$$D_{\mathrm{lm}} = \left(\frac{D_{\mathrm{v-a}}}{\phi_{\mathrm{ha}}}\right) \; . \tag{54}$$

The above results show that one should not view the time taken by the material point $F$ of water
while locked in the ice phase as slowing diffusion. Rather, the ice layer is an enormous reservoir of water
vapor, providing a continual source for diffusing water vapor until such time that the point $F$ reaches the
upper surface of the ice layer.
Finally, all of the results in this section were developed without reference to the hand-to-hand
mass transfer analogy. However, the hand-to-hand analogy provides an elegant shortcut to the identical
results of Eq. (54). This fact may be attributed to either: i) adopting the view of shortened diffusion paths,
or ii) adopting the view of an elevated intrinsic velocity as was done in Hansen (2019).
**6. Discussion**

This comment paper demonstrates that hand-to-hand water vapor transport provides an effective
model for correctly predicting enhanced diffusion in a layered ice/humid air microstructure. The model is
supported by rigorous control volume analyses using both a moving control volume and a fixed control
volume. Although the hand-to-hand concept is incredibly valuable, one can dispense with the hand-to-
diffusion mechanism and still achieve the same results of enhanced diffusion due to the "reservoir effect"
of the ice phase holding massive amounts of water vapor. In brief, ***the existing ice phase within the***
***layered microstructure is a major contributing factor to the overall mass transport of water moving***
***through the system.*** The approach of Fourteau et al. (2021a) ignores the contribution of mass diffusion
attributed to the reservoirs of water vapor contained within the ice layers.

The displacement time history seen in Figures 11 also demonstrates that there is no counterflux of
mass transfer due to a downward motion of the ice phase. Indeed, there is no negative motion of a
material point of water at any time, either in the ice phase or the humid air phase. While point $F$ is locked
in the ice phase with no motion, the ice phase steadily moves lower through deposition from water vapor
rising from below, producing the *appearance* of downward motion.
Let us now return to the known energy flux of the layered microstructure given by Eq. (9) and
repeated below as

$$q_{\mathrm{lm}} = -\left(\left(\frac{k_{\mathrm{ha}}}{\phi_{\mathrm{ha}}}\right) + \left(\frac{D_{\mathrm{v-a}}}{\phi_{\mathrm{ha}}}\right) u_{\mathrm{sg}} \frac{d\,\gamma_{\mathrm{v}}}{d\,\theta}\right) \frac{\partial\theta}{\partial x} \; . \tag{55}$$

The following observations can be made:
•    The thermal conductivity is given by
$$k_{\mathrm{lm}} = \left(\frac{k_{\mathrm{ha}}}{\phi_{\mathrm{ha}}}\right) \tag{56}$$

This expression is also precisely the thermal conductivity of humid air within the layered
microstructure as the energy flux of the layered microstructure is identical to the energy
flux of the humid air constituent.





- Consistent with a rigorous control volume analysis as well as the material point tracking analysis of Section 5.1, the diffusion coefficient is given by

$$D_{\mathrm{lm}} = \left(\frac{D_{\mathrm{v-a}}}{\phi_{\mathrm{ha}}}\right) \quad . \tag{57}$$

The above also represents the diffusion through the humid air constituent of the layered microstructure. In this sense, the decomposition of Eqs. (56) and (57) are identical to the results for humid air as a pure substance put forth by Bird et al. (1960) and outlined in Section 2.2.

A hand-to-hand model of water vapor transport produces the correct diffusion coefficient. Although the hand-to-hand description of Yosida (1955) is visually superb (outstanding in this writer's view), one could dispense with this concept in favor of the "reservoir effect" of water vapor transport. The reservoir effect has the desirable trait that the "nonphysical" nature of hand-to-hand water vapor diffusion is eliminated.

Equation (57) shows that diffusion in the layered microstructure is enhanced at all volume fractions compared to diffusion in humid air as a pure substance. The sublimation and deposition of water vapor across ice grains in snow is also clearly present during temperature gradient metamorphism of snow as it is leads to microstructural evolution. ***Hence, it is entirely possible, indeed probable, for macroscopic water vapor diffusion to be enhanced in snow compared to diffusion in humid air as a pure substance.*** An analysis suggesting this diffusion enhancement was provided in Hansen (2019). Present work by the author suggests that for an ice volume fraction of 0.3, the normalized diffusion coefficient for snow ranges from approximately 0.9-1.3, depending on the degree to which hand-to-hand mass transport is present.

Efforts to quantify precise values of the diffusion coefficient have been limited by confusion over the definition of this important parameter. The present paper provides the clarity to move forward in a consistent manner.

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

**Nomenclature**

948        **Latin Letters**
$D$        diffusion coefficient
$k$        thermal conductivity
$\boldsymbol{n}$        unit normal
$q$        energy flux
$t$        time
$u_{sg}$    latent heat of sublimation of ice
$v$        velocity
$x$        macroscale coordinate

959        **Greek Symbols**
$\xi$        microscale coordinate
$\gamma_v$    density of vapor component
$\rho$        density
$\theta$    absolute temperature
$\phi_\alpha$    volume fraction of constituent $\alpha$

966        **Superscripts**
(c)        conduction





| 968 | (d) | diffusion |
| 969 | | |
| 970 | | **Subscripts** |
| 971 | C | reference frame moving with ice front |
| 972 | f | advancing ice front due to ice accretion |
| 973 | i | ice constituent |
| 974 | ha | humid air constituent |
| 975 | lm | layered microstructure |
| 976 | v | vapor component within humid air |
| 977 | v-a | water vapor in air |
| 978 | s | snow |
| 979 | | |
| 980 | | |





**List of Figures**



**Figure 1.** Water vapor transport mechanisms in snow exhibiting: a) hand-to-hand water vapor transport (red arrows) involving continuous sublimation and condensation of water vapor and, b) water vapor passing around ice grains (green arrows).

**Figure 2.** (a) Layered ice/humid air microstructure shown in green bounded by solid ice blocks. (b) A macroscale continuum point of the layered microstructure showing the presence of both ice and humid layers.

**Figure 3.** Arbitrary fixed region $\mathcal{R}$ showing a surface flux of the vector field, $\boldsymbol{F}$, leaving the boundary defined by the surface, $\partial\mathcal{R}$.

**Figure 5.** (a) Layered microstructure shown in green bounded by solid ice *blocks*. (a) Coincident fixed and moving control volumes at time $t = 0$, shown by the dashed line. (b) Fixed control volume (red) and moving control volume (green) advancing with the ice front at time $t$.

**Figure 6.** (a) Humid air as a pure substance under a temperature gradient showing an advancing ice front of length $\ell$ occurring in a characteristic time $\tau$. (b) A layered microstructure with ice volume fraction $\phi_i = 1/3$ showing an advancing ice front of length $1.5\,\ell$ occurring in the same characteristic time $\tau$ as for humid air alone shown in (a)—see the discussion in Section 4.2.

**Figure 7.** (a) Unit cell with a moving control volume that advances downward with the moving ice front cause by ice accumulation on the lower surface of the ice constituent. (b) Alternate configuration of the unit cell where the moving control volume extends through the middle of the ice phase.

**Figure 8.** Fixed control volume showing a unit cell with: (a) the ice phase at time $t = 0$, and (b) the ice phase at a later time, $t$.

**Figure 9.** (a) Fixed control volume of characteristic length $1.5\ell$, filling with the advancing ice front in time $\tau$, (b) Fixed control volume enclosing the lower solid ice/ice mixture boundary where sublimation is occurring.

**Figure 11.** Motion of a continuum material point, $F$, of water as it traverses through humid air, is locked within an ice layer, and then is released via sublimation to again traverse through humid air to reach the upper boundary of solid ice.

fig01

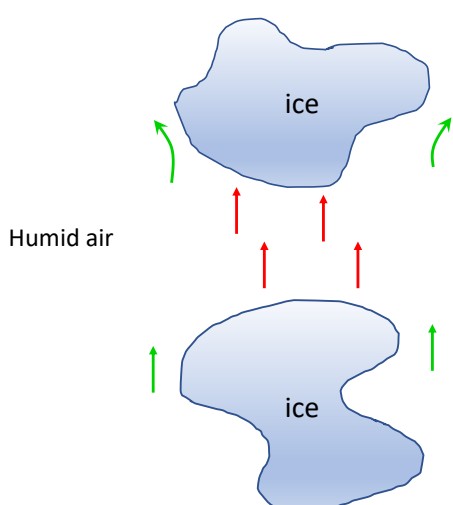



fig02

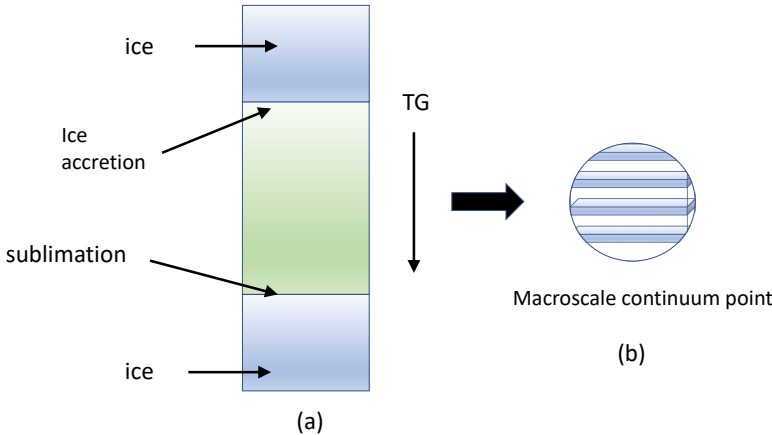



fig03

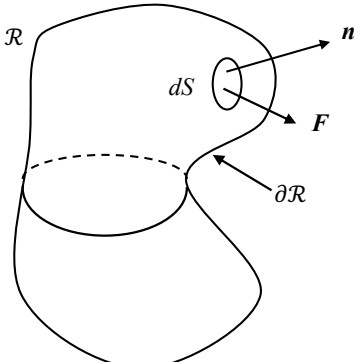





fig04

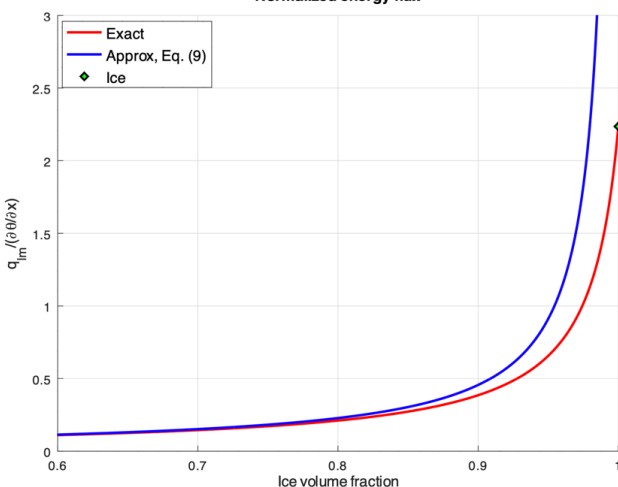



fig05

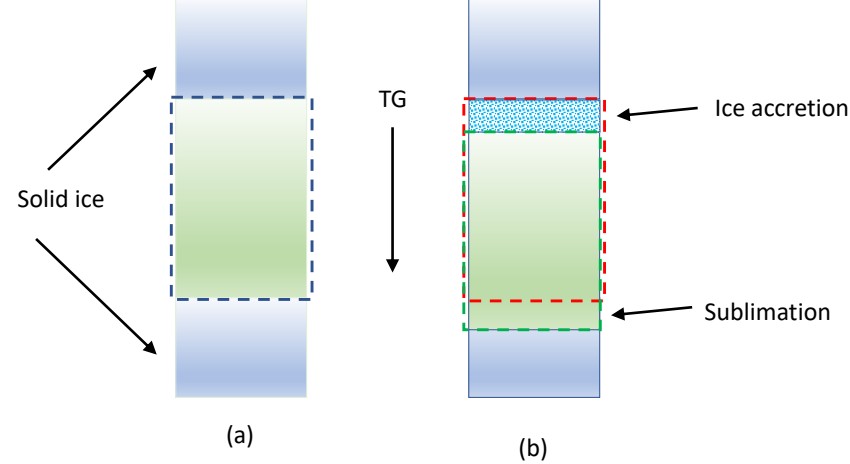



fig06

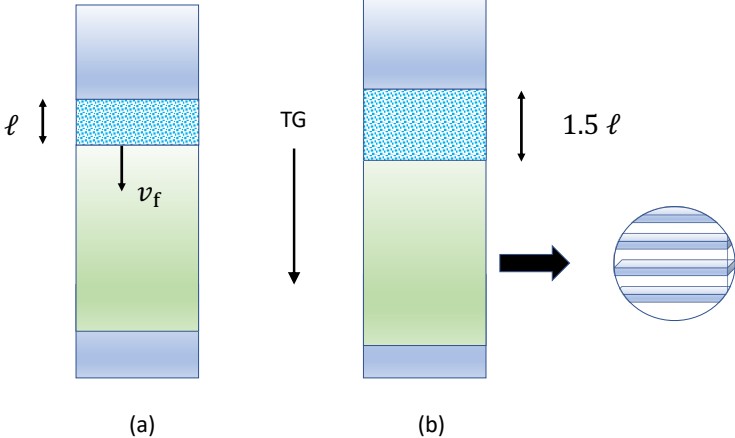



fig07

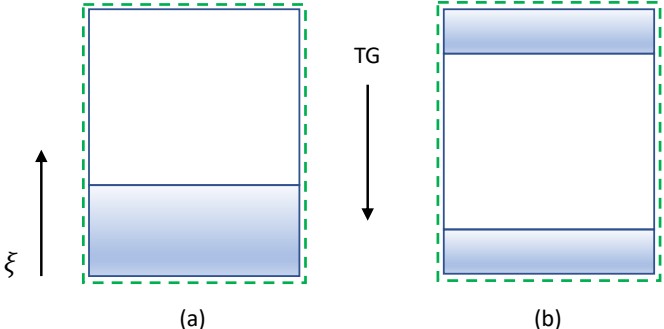

(a)        (b)



fig08

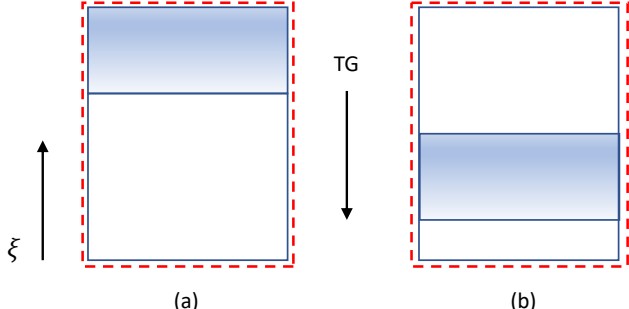

fig09

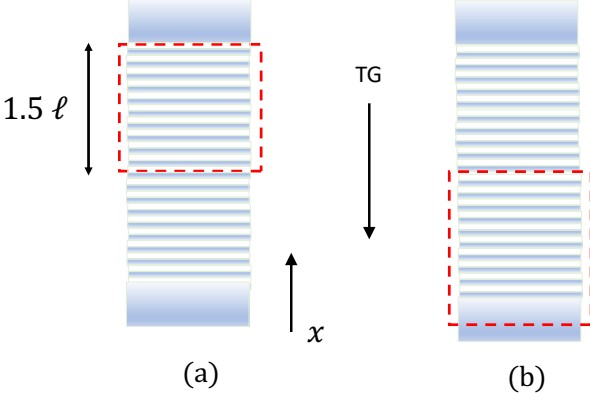

(a)  (b)



fig10

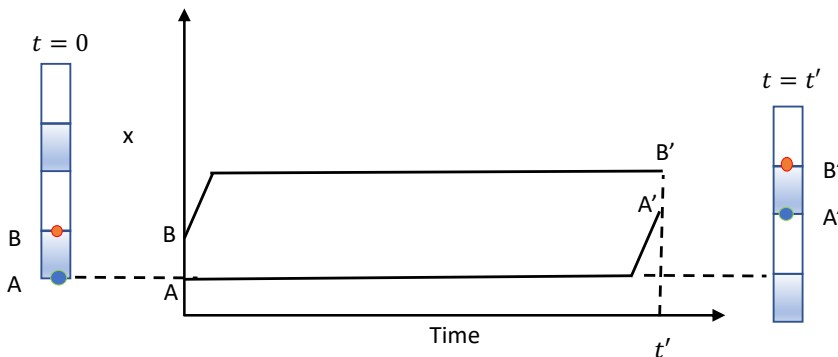





fig11

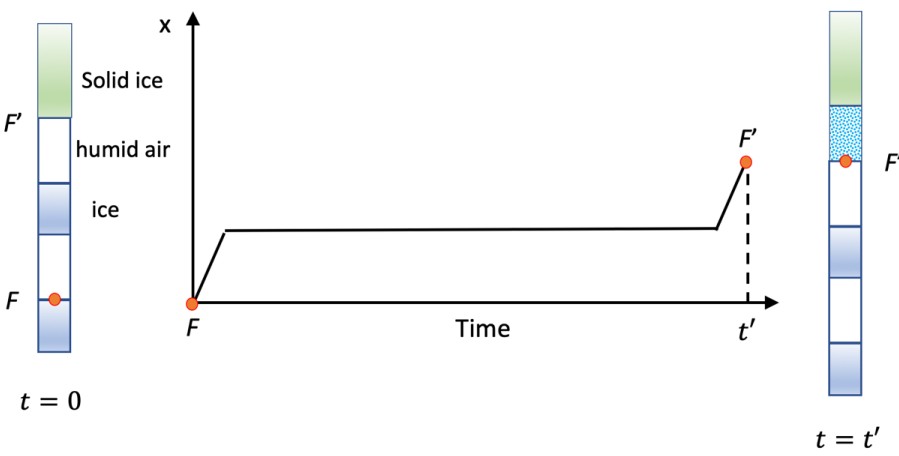