# Peer review of "Macroscopic water vapor diffusion is not enhanced in snow"

_The Cryosphere, 2022_

## Referee Comment (RC1)

**Review on Comment on: Macroscopic water vapor diffusion is not enhanced in snow**

Dear Editor,

This *comment paper* discusses how the work of Fourteau et al. is supposedly incorrect. Since I have reviewed and approved this particular paper, I owe it to the writer of the comment and the writers of paper to at least take a look at the major issues that the writer of the *comment* brings to the table. In conclusion that is brought forward by my review below I found the critics to be unfounded. In the following the I use *comment* to address the comment paper.

As is clear from the discussion section both sides claim the other one is wrong with varying degrees of proof, and leap on a tangent expressing their own views on the matter. This stands at the origin, as the historical discussion already suggests, of an exponentially growing amount of discouraging work of fact checking. My time dedicated to the matter is purely based on the respect both authors deserve.

Firs of all, this is my *opinion*, take it or leave it: Please try to be civil. Any subjective adjectives and expressions of feelings should have no place here, especially in a public place of scientific debate: keep the mud-throwing for an obscure party at a soon to be forgotten conference. Whether you are aware of it or not, this makes for a very unpleasant working environment, and it should not come at a surprise that sincere scientists who do not like conflict, but love a good scientific debate are likely to flee the scene.

To the grid of of vapor diffusion. I will comment only on the critics raised by the *comment* on the paper, and not on content that is tangent to the paper. If a new paper on vapor flux is submitted, I will consider reviewing it by that point.

Please note that in order to conclude on this matter it might be wise to seek a third reviewer that has no previous ties (writer, editor, reviewer of, professional relation) to either Fourteau et al. or Hansen.

I will first comment on the first half of the abstract and then comment on the issues raised by the *comment*'s' author in the main body of the text.

**abstract**

The abstract in red, my comments in black.

The central thesis of the authors paper is that macroscopic water vapor diffusion is not enhanced in snow compared to diffusion through humid air alone. Further, mass diffusion occurs entirely as the result of water vapor diffusion in the humid air at the microscale and the ice phase has no effect other than occupying volume where diffusion cannot occur.

Technically this is not true. The paper deals with the ice phase appropriately. The ice phase acts as sinks and sources to balance the diffusion present in the humid air. This is enforced by the Robin boundary condition (Eq. 3)

The foundation of their conclusion relies on the premise that the synchronous sublimation and deposition of water vapor across ice grains, known as hand-to-hand water vapor transport, does not lead to enhanced mass diffusion.

Synchronous sublimation and deposition of water vapor across ice grains is not "known" as hand-to-hand vapor transport. Some authors have used the term to explain mass transfer found in experiments that suggest faster transport than diffusion of water vapor alone. There are other candidates, such as convection, wind pumping and

experimentally flawed definitions that could explain the matter too. The authors of the accused paper do not make any assumptions on transport mechanisms. They 'just' perform a numerical experiment that show that when transport is purely diffusion limited, it does not speed up mass transport of water vapor.

We use a layered microstructure to rigorously show that diffusion is enhanced at all ice volume fractions compared to diffusion through humid air alone, and further, the hand-to-hand model of diffusion correctly predicts this diffusion enhancement.

This sentence is technically incorrect, since there is no water vapor diffusion in the ice volume fractions, simply because it is a difference phase and now molecules are moving. The second part of the sentence relates to a prediction of a diffusion enhancement. I haven't found an experiment to date that actually shows that the predicted model based on enhanced transport in the layered microstructure is actually correct.

The authors attempt to dismiss the concept of enhanced mass transfer resulting from hand-to-hand water vapor transport by arguing that there is a counterflux of water vapor in the form of downward motion of ice.

This is not what the authors do, the base their conclusion on observations of their numerical simulations based on well defined set of equations and boundary conditions.

While the ice phase appears to be propagating downward, all continuum material points of water (either vapor or ice) are moving upward (counter to the temperature gradient) with a monotonically increasing (nonnegative) motion.

Monotonically increasing motion? What is the cause of the increase of motion? Within the context of this discussion, the flux of vapor is assumed to be uniform over the control volume.

Specifically, material points of water in vapor form are diffusing upward through the humid air while material points of water in the form of ice are at zero velocity while locked in the ice phase. Material points of water never exhibit downward motion, despite the ice phase appearance of downward motion. Since the motion of all material points of water is monotonically increasing for all time, there is no counter flux of mass due to downward motion of the ice and such apparent motion is a mirage in the context of mass transfer.

Again, what is monotonically increased motion? What is mend by mirage? The authors have argued at some point, that if you do not include the ice phase in the control volume one speaks about an apparent flux (Hansen). And as a consequence there are other 'apparent' fluxes that you must calculate that to get the correct answer. This has been adequately explained by the rebuttal of the authors of the paper.

The rest of the abstract addresses content which is how vapor diffusion should be defined in the eyes of the author. I will not go into this at this point.

**General comments:**

As I understand from the comment and the discussion, the author criticized that the paper on mainly two points. The first being that they have used the volume averaging technique wrongly and secondly about the decomposition of the heat flux into a conduction and vapor diffusion part. I will discuss these two points further below.

**The first point**

In the discussion he takes the reviewers Fig. 1 as an example that highlights the hart of the problem. He states that by taking an arbitrary control volume that includes the

ice phase and is not moving with the ice sublimation/deposition interface velocity the paper neglects the temporal nature of the problem. I will argue why this is not the case.

Volume averaging can always be applied, but the outcome is either a local or macroscopic measure depending on whether the integrated volume represents a the macroscopic microstructure, i.e. a statistical representative volume. By introducing a toy-model such as a layered microstructure a unit-cell (with periodicity in its geometry) will suffice, since having many copies of them will not change the outcome. But there are rules for unit-cells that are often violated by the author of the *comment*. Not including the zero contribution in the ice phase will not lead to a zero contribution to the average, but will lead to a wrong normalization factor in the denominator $\mathcal{V} = \mathcal{V}_i + \mathcal{V}_{h-a}$:

$$F = \overline{f} = \frac{1}{\mathcal{V}} \int_{\mathcal{V}} f \, dv \tag{1}$$

Although the temperature gradient is enhanced by approximately $1/\phi$ with $\phi$ the porosity, at most as shown later, it is averaged out when correctly including a zero contribution of the ice phase.

All authors agree that the *local* diffusion in the humid air is enhanced because the local temperature gradient, which is forcing the diffusion, is enhanced, but they differ in how to compute its macroscopic average.

The author of the comment suggest that the paper fails to move to a co-moving frame, the natural frame of the problem. I don't share this opinion, since there are no actual material points moving. It is rather a question of sampling enough of the microstructure such that you get a reliable macroscopic flux. If a static frame is chosen to be a unit cell of the layered microstructure (including the ice phase!) there is never a change in density within this cell, hence no need to apply Taylors averaging theorem. By volume averaging the unit cell it is naturally sampling all fluxes through all possible surfaces at all time stages of the cell, since each slice,(all together building up the total volume of the unit cell), is representing any slice at a certain time. In other words: take one cross-section in either ice or humid air and evaluate its flux at all times until the ice matrix has moved a unit-length of the unit cell, is exactly the same as volume averaging the unit cell. Therefore there is no need to move into the a *co-moving* unit frame. Note that if you are in a moving frame, you could take the boundaries also through the ice phase instead of the lying on the interface and by the same argument given by the author of the *comment* in the rebuttal of Reviewer 1, it mass flux is equal to the mass balance given at the moving interface, regardless of the thickness of the humid air phase. This will lead to the conclusion that the enhancement is independent of porosity ... which is a dubious outcome regardless. This also means that there is mass transport through an impermeable ice layer in the midst of a snow pack, by means of diffusion.

There is also no disagreement on the mass conservation between the flux in the humid air and deposition/sublimation rates. These are correctly identified in the paper [Fourteau et al.] Eq. (2).

Therefore I don't see any legitimate points of criticism in the volumetric averaging approach used by Fourteau et al..

**The second point**

When we describe mass flux of vapor diffusion it only makes sense to start from the local vapor flux and upscale to the macroscopic description with an appropriate method. Since we assume (ground truth) saturated conditions $\gamma_v(x) = \gamma_{v,\text{sat}}(\theta(x))$, it

becomes mathematically the same problem as the heat equation, meaning: once we have established the temperature field at all locations we also know the vapor field and its diffusive fluxes. However, starting from the solution of those equations and assigning its contributions to conductive fluxes, vapor fluxes and latent heat fluxes is not straightforward. As argued by the rebuttal to this *comment*, previously proven by the reviewer of Hansen and Foslien, and identified in the Appendix B of Fourteau et al., the decomposition into conductive and effective diffusion as suggested by Hansen and Foslien serves as definition of the effective diffusion constant and has yet to be (dis)proven to be equivalent to the volume averaging approach. However it is not upon the authors of Fourteau et al. to disprove this, it is up to the author of Hansen and Foslien to prove that it is equivalent, since volume averaging is a legitimate method of calculating effective transport properties of porous media Whitaker, Torquato.

Since we have established that volume averaging, as done by Fourteau et al. is done correctly, the only part that one can criticize in their paper is that they haven't included the latent heat when solving the heat equation to obtain $\theta(\mathbf{x})$. The authors [Fourteau et al.] have argued that this would lead to a decrease in the temperature gradients in the pore-space and can therefore not lead to any enhanced fluxes (Section 5, first paragraph). The question is therefore essentially this: is this true or not? They more or less show that in their Appendix C, but it is not that straight forward to me.

In the following I will proof to myself that this is true. Note that this is a tangent line of thoughts too. Therefore feel free to ignore it.

In case of the laminar microstructure it is easy to solve the heat equation. What we are after is to evaluate to what extent the temperature gradient in humid air is enhanced and to what extent it depends on latent heat flux in the ice phase (the phase transitions).

To start from ground truths, with macroscopic temperature gradient $G$, expressed in volume averaged temperature gradients in the humid air (h-a) with porosity $\phi$, and ice with volumetric density $1 - \phi$

$$G = \frac{\Delta\theta}{\Delta x} = \phi \left(\frac{d\theta}{dx}\right)_{\mathrm{h-a}} + (1 - \phi) \left(\frac{d\theta}{dx}\right)_{\mathrm{i}} \tag{2}$$

Given the local boundary condition (for solving the heat equation $\nabla^2\theta = 0$) at the depositing interface

$$k_i \left(\frac{d\theta}{dx}\right)_i - k_{\mathrm{h-a}} \left(\frac{d\theta}{dx}\right)_{\mathrm{h-a}} = L_{\mathrm{sg}} w_n, \tag{3}$$

with all gradients $> 0$. Here the latent heat is given by $L$sg and the velocity of the depositing interface $w_n > 0$ and conductivity of ice $k_{\mathrm{i}}$ and humid air $k_{\mathrm{h-a}}$. For the sublimating side all terms are multiplied by -1 because the normal vector changes sign (see Eq.9 in Calonne et al.).

For fixed gradient $G$ over the unit cell (however you want to define it, given the periodicity requirement) we can express the $G$ in terms of the local temperature gradient and latent heat contributions, using the previously stated boundary condition:

$$G = \phi \left(\frac{d\theta}{dx}\right)_{\mathrm{h-a}} + (1 - \phi) \left[\frac{k_{\mathrm{h-a}}}{k_{\mathrm{i}}} \left(\frac{d\theta}{dx}\right)_{\mathrm{h-a}}\right] + (1 - \phi)\frac{L_{\mathrm{sg}} w_n}{k_{\mathrm{i}}} \tag{4}$$

$$= \left(\frac{d\theta}{dx}\right)_{\mathrm{h-a}} \left[\phi + (1 - \phi)\frac{k_{\mathrm{h-a}}}{k_{\mathrm{i}}}\right] + (1 - \phi)\frac{L_{\mathrm{sg}} w_n}{k_{\mathrm{i}}} \tag{5}$$

Therefore the we can write

$$\left(\frac{d\theta}{dx}\right)_{\mathrm{h-a}} = \left(G - (1-\phi)\frac{L_{\mathrm{sg}}w_n}{k_{\mathrm{i}}}\right) \bigg/ \left[\phi + (1-\phi)\frac{k_{\mathrm{h-a}}}{k_{\mathrm{i}}}\right] < G/\phi \qquad (6)$$

Note that all terms in the right-hand-side of the equation are constants. From this we can conclude that the latent heat contribution can only reduce the local gradient in humid air. By setting the latent heat to zero (the contribution in this case is less than $10^{-3}$ compared to the temperature gradient which is usually in the order of $10^1$) and acknowledging the $\frac{k_{\mathrm{h-a}}}{k_{\mathrm{i}}} \approx 10^{-2}$, the local gradient $\left(\frac{d\theta}{dx}\right)_{\mathrm{h-a}} < G/\theta$. When we use this upper bound to calculate the effective diffusion constant by volume averaging the vapor flux we obtain

$$F \equiv \qquad D_{\mathrm{eff}}\frac{d\gamma_v}{d\theta}G \qquad (7)$$

$$= \quad \frac{D_{\mathrm{v-a}}}{\mathcal{V}} \int_{\mathcal{V}} \frac{d\gamma_v}{d\theta}\frac{d\theta}{dx}\, dv \qquad (8)$$

$$= \quad \frac{D_{\mathrm{v-a}}\phi}{\phi+(1-\phi)}\frac{d\gamma_v}{d\theta}\left(\frac{d\theta}{dx}\right)_{\mathrm{h-a}} \qquad (9)$$

$$< \qquad D_{\mathrm{v-a}}\frac{d\gamma_v}{d\theta}G \qquad (10)$$

Therefore we can conclude that $D_{\mathrm{eff}} < D_{\mathrm{d-a}}$ and concludes the matter.

**References**

N. Calonne, C. Geindreau, and F. Flin. Macroscopic modeling for heat and water vapor transfer in dry snow by homogenization. 118(47):13393–13403. doi: 10.1021/jp5052535.

K. Fourteau, F. Domine, and P. Hagenmuller. Macroscopic water vapor diffusion is not enhanced in snow. URL https://tc.copernicus.org/preprints/tc-2020-183/.

A. Hansen. Revisiting the vapor diffusion coefficient in dry snow. pages 1–27. ISSN 1994-0416. doi: https://doi.org/10.5194/tc-2019-143. URL https://www.the-cryosphere-discuss.net/tc-2019-143/#discussion.

A. C. Hansen and W. E. Foslien. A macroscale mixture theory analysis of deposition and sublimation rates during heat and mass transfer in dry snow. 9(5):1857–1878. ISSN 1994-0424. doi: 10.5194/tc-9-1857-2015. URL https://tc.copernicus.org/articles/9/1857/2015/.

S. Torquato. *Random heterogeneous materials: microstructure and macroscopic properties.* Number 16 in Interdisciplinary applied mathematics. Springer. ISBN 978-1-4757-6357-7 978-0-387-95167-6 978-1-4757-6355-3. OCLC: 248518949.

S. Whitaker. *The Method of Volume Averaging.* Theory and Applications of Transport in Porous Media. Springer Netherlands. ISBN 978-0-7923-5486-4. URL https://books.google.ch/books?id=x7mQCEokSCAC.

---

## Referee Comment (RC2)

**Review of**
**"Comment on: Macroscopic water vapor diffusion is not enhanced in snow"**
**by Andrew C. Hansen**

To simplify the reading, I will use Hansen's notations.

**Two types of average**

Assuming the existence of a Representative Elementary Volume $Y = Y_{ha} \cup Y_i$ where $Y_{ha}$ and $Y_i$ are respectively the volume occupied by the humid air and the ice phase, two types of average must be distinguished:

- the $\alpha$-phase intrinsic average ($\alpha \in \{ha, i\}$)

$$\langle \bullet \rangle^\alpha = \frac{1}{|Y_\alpha|} \int_{Y_\alpha} \bullet \, dV \tag{i}$$

- and the volume average

$$\langle \bullet \rangle = \frac{1}{|Y|} \int_Y \bullet \, dV = \phi_{ha} \langle \bullet \rangle^{ha} + (1 - \phi_{ha}) \langle \bullet \rangle^i \tag{ii}$$

The paradox raised by Hansen comes from the difference between these two averages for the water vapor flow $\mathbf{J}_v$. As the water vapor flux is 0 in the ice phase

$$\langle \mathbf{J}_v \rangle = \phi_{ha} \langle \mathbf{J}_v \rangle^{ha} \tag{iii}$$

**Homogenization in the general case**

More than 30 years ago I worked on the heat and mass transfer in porous wet media [1, 8, 2]. If the solid surfaces of the porous material are wet, confounding the solid and liquid thermal conductivities, which are now the ice conductivity, the problem is exactly the same as the heat and mass transfer in snow. The case of wet porous media is easier to investigate experimentally measuring the macroscopic heat conductivity. More increasing the temperature leads to an increase of the contribution of the water vapor transport to the heat transfer helping to elucidate clearly the mechanisms.

We do both experiments [1] and the homogenization of the process using the volume averaging technique [8]. The main conclusions of this theoretical part are briefly recalled. The technical details are given in reference [2] unfortunately in french.

Two equations have to be solved at the snow macroscale corresponding to the water vapor mass balance and the heat transfer equation:

$$\phi_{ha} \frac{\partial \gamma_v}{\partial t} = \boldsymbol{\nabla} \cdot \left( \frac{d\gamma_v}{d\theta} \frac{\boldsymbol{D}}{1 - x_v} \cdot \boldsymbol{\nabla}\theta \right) - \dot{m} \tag{iva}$$

$$\langle \rho c \rangle \frac{\partial \theta}{\partial t} = \boldsymbol{\nabla} \cdot (\boldsymbol{k} \cdot \boldsymbol{\nabla}\theta) + u_{sg} \dot{m} \tag{ivb}$$

where

$\boldsymbol{k}$     is the Fourier thermal conductivity of the medium defined at the snow scale,

$\boldsymbol{\nabla}\theta$     $\equiv \langle \boldsymbol{\nabla}\theta \rangle$ is the volume average temperature gradient,

$\boldsymbol{D}$     is the macroscopic water vapor diffusion coefficient,

$\langle \rho c \rangle$     $= \phi_{ha}(\rho c)_{ha} + \phi_i (\rho c)_i$ is the volume heat capacity of the snow,

$\dot{m}$     is the mass of condensed water vapor into ice per unit volume of snow,

$u_{sg}$     the latent heat of sublimation.

In equation (iva) for the the vapor transport, note that the volume average of the water vapor flux is $\langle \mathbf{J}_v \rangle = -\dfrac{\mathrm{d}\gamma_v}{\mathrm{d}\theta} \dfrac{\boldsymbol{D}}{1 - x_v} \cdot \boldsymbol{\nabla}\theta$, which defines the vapor diffusion coefficient (tensor) $\boldsymbol{D}$ at the macroscale for the snow. The factor $1 - x_v$, where $x_v$ is the vapor mole fraction in humid air in the denominator introduced by assuming stagnant air, can be discarded in the case of snow as the mole fraction $x_v$ is small.

Combining the two preceding relations with the reasonable hypothesis that the transport and phase change terms are much larger than the temporal variation of the water vapor density leads to

$$\langle \rho c \rangle \frac{\partial \theta}{\partial t} = \boldsymbol{\nabla} \cdot (\boldsymbol{k}_{app} \cdot \boldsymbol{\nabla}\theta) \qquad \text{(v)}$$

where the *apparent* thermal conductivity $\boldsymbol{k}_{app}$ is given by

$$\boldsymbol{k}_{app} = \boldsymbol{k} + u_{sg} \frac{\mathrm{d}\gamma_v}{\mathrm{d}\theta} \boldsymbol{D} = \boldsymbol{k} + k_{dif} \frac{\boldsymbol{D}}{\mathcal{D}_{v-a}} \qquad \text{(vi)}$$

with $k_{dif} = u_{sg} \mathcal{D}_{v-a} \dfrac{\mathrm{d}\gamma_v}{\mathrm{d}\theta}$. The closure problem to be solved to determine the macroscopic coefficients is given for completeness in the Appendix. Starting from standard equations at the microscale the asymptotic homogenization process is a purely mathematical tool without any additional physical hypothesis. The homogenization process is similar to reference [3] except for the condition at the ice surface where thermodynamic equilibrium is imposed (air is saturated, which is called *fast kinetics hypothesis* in [5, 6]) rather than the impervious condition imposed by [3].

The normalized water vapor diffusion coefficient defined as $\mathbf{f} = \dfrac{\boldsymbol{D}}{\mathcal{D}_{v-a}}$ verifies the relation (see [2] and Appendix)

$$\phi_{ha} \langle \boldsymbol{\nabla}\theta \rangle^{ha} = \mathbf{f} \cdot \left( \phi_{ha} \langle \boldsymbol{\nabla}\theta \rangle^{ha} + \phi_i \langle \boldsymbol{\nabla}\theta \rangle^i \right) \qquad \text{(vii)}$$

which proves, at least intuitively, that in the isotropic case $f$ is less than 1 and that therefore the solid ice phase acts as a resistance to water vapor transport.

**Historical note**

The problem of simultaneous heat and mass transfer in wet porous media goes back to Krischer [7], who pointed out the important contribution of the evaporation-condensation process to the thermal conductivity noting that at temperature larger than 65 ºC, $k_{dif} > k_{water}$. Therefore increasing water saturation, that is replacing $k_{dif}$ by $k_{water}$, leads to a decrease of the apparent thermal conductivity. In the soil domain, the first contributors are Philip and de Vries [9] and de Vries [4]. Interestingly the unnecessary idea of an enhanced water vapor diffusion in porous media also exits in the soil domain: the fluid phase is supposed to transfer immediately the condensing water vapor due to a modification of the curvature of the meniscus (see figure 2 in [9]).

**Layered microstructue**

Application of homogenization to the particular case of a series medium (layered microstructure) has been considered in reference [2]:

$$k_{lm} = \frac{k_i \left(k_{ha} + (1 - \phi_{ha}) k_{dif}\right)}{(1 - \phi_{ha}) \left(k_{ha} + k_{dif}\right) + \phi_{ha} k_i} \tag{viiia}$$

$$f = \frac{D_{lm}}{\mathcal{D}_{v-a}} = \frac{\dfrac{\phi_{ha}}{k_{ha} + k_{dif}}}{\dfrac{1 - \phi_{ha}}{k_i} + \dfrac{\phi_{ha}}{k_{ha} + k_{dif}}} \leq 1 \tag{viiib}$$

$$k_{app} = k_{lm} + k_{dif}\frac{D_{lm}}{\mathcal{D}_{v-a}} = \frac{1}{\dfrac{1 - \phi_{ha}}{k_i} + \dfrac{\phi_{ha}}{k_{ha} + k_{dif}}} \tag{viiic}$$

The preceding result can be proved using simple arguments. First as the equivalent conductivity of the humid air phase is $k_{ha} + k_{dif}$, the apparent conductivity $k_{app}$ of the layered medium is straightforwardly given by equation (viiic). Therefore the temperature gradient in the humid air is given by

$$(k_{ha} + k_{dif}) \left\langle \frac{d\theta}{dx} \right\rangle^{ha} = \frac{1}{\dfrac{1 - \phi_{ha}}{k_i} + \dfrac{\phi_{ha}}{k_{ha} + k_{dif}}} \left\langle \frac{d\theta}{dx} \right\rangle \tag{ix}$$

The water vapor flux in the humid air phase $\langle J_v \rangle^{ha}$ is given by

$$\langle J_v \rangle^{ha} = -\mathcal{D}_{v-a} \frac{\dfrac{1}{k_{ha} + k_{dif}}}{\dfrac{1 - \phi_{ha}}{k_i} + \dfrac{\phi_{ha}}{k_{ha} + k_{dif}}} \left\langle \frac{d\theta}{dx} \right\rangle \tag{x}$$

and the volume average water vapor flux is therefore given by

$$\langle J_v \rangle = \phi_{ha} \langle J_v \rangle^{ha} + (1 - \phi_{ha}) \langle J_v \rangle^{i} = \phi_{ha} \langle J_v \rangle^{ha} \tag{xi}$$

as the water vapor flux is 0 in the ice phase. Finally the normalized water vapor diffusion coefficient is given by

$$f = \frac{D_{lm}}{\mathcal{D}_{v-a}} = \frac{\dfrac{\phi_{ha}}{k_{ha} + k_{dif}}}{\dfrac{1 - \phi_{ha}}{k_i} + \dfrac{\phi_{ha}}{k_{ha} + k_{dif}}} \tag{xii}$$

The expression (viiia) for $k_{lm}$ is simply deduced from the relation $k_{app} = k_{lm} + k_{dif}\dfrac{D_{lm}}{\mathcal{D}_{v-a}}$.

If the solid (ice) conductivity is much larger than the gas or the diffusive conductivity as

considered by Hansen, we have:

$$k_{lm} = \frac{k_{ha} + (1 - \phi_{ha})\,k_{dif}}{\phi_{ha}} \tag{xiiia}$$

$$f = \frac{D_{lm}}{\mathcal{D}_{v-a}} = 1 \tag{xiiib}$$

$$k_{app} = \frac{k_{ha} + k_{dif}}{\phi_{ha}} = \frac{k_{ha} + u_{sg}\mathcal{D}_{v-a}\dfrac{\mathrm{d}\gamma_v}{\mathrm{d}\theta}}{\phi_{ha}} \tag{xiiic}$$

$$\tag{xiiid}$$

These results are exactly those of [5, 6]. If in the decomposition of $k_{app}$, the heat conductivity only due to conduction $k_0$ is introduced

$$k_{app} = k_0 + f_{exp}\,k_{dif} \quad \text{with } k_0 = \frac{1}{\dfrac{\phi_{ha}}{k_{ha}} + \dfrac{(1 - \phi_{ha})}{k_i}} \tag{xiv}$$

where the so-called experimental diffusion water diffusion factor $f_{exp}$ (because easy to determine experimentally) is written as [1, 8]

$$f_{exp} = \frac{1}{\phi_{ha}\left(1 + \dfrac{\phi_i k_{ha}}{\phi_{ha} k_i}\right)\left(1 + \dfrac{\phi_{hi}(k_{ha} + k_{dif})}{\phi_{ha} k_i}\right)} \tag{xv}$$

It is immediate to check that $f_{exp}$ can exceed unity. In the limiting case of a very large value of $k_i$, i.e. when $(1 - \phi_{ha})/k_i \ll \phi_{ha}/(k_{ha} + k_{dif})$, the maximum value obtained is $f_{exp} \to 1/\phi_{ha}$.

These findings agree fully with Fourteau et al. [5, 6].

**Comparison with Hansen's result**

The table below summarizes Hansen's results and the result of the homogenization calculation.

| | Hansen | Homogenization |
|---|---|---|
| $k_{lm}$ | $\dfrac{k_{ha}}{\phi_{ha}}$ | $\dfrac{k_{ha} + (1 - \phi_{ha})\mathcal{D}_{v-a}u_{sg}\dfrac{\mathrm{d}\gamma_v}{\mathrm{d}\theta}}{\phi_{ha}}$ |
| $D_{lm}$ | $\dfrac{\mathcal{D}_{v-a}}{\phi_{ha}}$ | $\mathcal{D}_{v-a}$ |
| $k_{app} = k_{lm} + D_{lm}u_{sg}\dfrac{\mathrm{d}\gamma_v}{\mathrm{d}\theta}$ | $\dfrac{k_{ha} + \mathcal{D}_{v-a}u_{sg}\dfrac{\mathrm{d}\gamma_v}{\mathrm{d}\theta}}{\phi_{ha}}$ | idem |

If the definition (11) proposed by Hansen of the energy flux is correct, the splitting between the conductive flux (12) and the mass flux (13) does not give the rigorous result furnished by the homogenization procedure.

**Definition of the macroscopic diffusion coefficient or the thermal conductivity**

It should be noted that the definition of the macroscopic diffusion coefficient or the thermal conductivity has nothing to do with the modification of the snow structure. The definition of these coefficients implies that, when subjected to a concentration or a temperature gradient, the medium reacts *instantaneously* by a mass or heat flux. Since the ice front velocity is much lower than the vapor velocity, the medium can be assumed motionless. The temporal rearranging of the snow is another problem.

**Other comments**

• Line 299-303
The diffusion coefficient (16) obtained also by Fourteau et al. [5] is defined to represent the volume average water vapor flux $\langle \mathbf{J}_v \rangle$, which is needed in the macroscopic equations (iv). The flux leaving the upper boundary condition is a microscopic flux in the humid air phase, which in the particular case of the layered medium is equal to the humid air phase average: $\langle \mathbf{J}_v \rangle^{ha} = \langle \mathbf{J}_v \rangle / \phi_{ha}$.

• Line 305-309
The Fourier thermal conductivity $k_{lm}$ defined by (xiiia) obviously contains a *diffusive part*. If not, using $k_{app} = k_0 + f_{exp}\lambda_{dif}$ can lead to $f_{exp}$ values larger than 1, which is not expected.

• §3.4
At the homogenized macroscale, it is no more possible to distinguish separately the contribution of the two phases. This is evident when thinking about more complex geometries than the layered microstructure.

**Conclusion**

I completely agree with the calculations and arguments of Fourteau et al. [5, 6]. Hansen's calculations in the simple case of a multilayer structure are quite correct considering the internal modification of the snow structure. But his definition of the macroscopic coefficients (the splitting between the conductive and the diffusive parts) does not coincide with the precise rules given by the scaling process.

**References**

[1] S. Azizi, C. Moyne, and A. Degiovanni. Approche expérimentale et théorique de la conductivité thermique des milieux poreux humides. I. Expérimentation. *Int. J. Heat Mass Transfer*, 31(11):2305-2317, 1988.

[2] S. Azizi, J.C. Batsale, A. Degiovanni, and C. Moyne. Reply to "Discussion of "Approche expérimentale et théorique de la conductivité thermique des milieux poreux humides", Letter to editor. *Int. J. Heat Mass Transfer*, 33(8):1778-1779, 1990.

[3] N. Calonne, C. Geindreau, F. Flin. Macroscopic modeling for heat and water vapor transfer in dry snow by homogenization. J. Phys. Chem. B. 118:13393-13403, 2014.

[4] D. A. De Vries. Simultaneous transfer of heat and moisture in porous media. Trans. Am. Geophys. Un., 39:909-916, 1958.

[5] K. Fourteau, F. Domine, and P. Hagenmuller. Macroscopic water vapor diffusion is not enhanced in snow? Preprint. Discussion, The Cryosphere, https://doi.org/105194/tc-2020-183, 2021.

[6] K. Fourteau, F. Domine, and P. Hagenmuller. Impact of water vapor diffusion and latent heat on the effective thermal conductivity of snow. The Cryosphere, 15:2739-2755, https://doi.org/105194/tc-15-2739-2021, 2021.

[7] 0. Krischer. Die wissenschaftlichen Grundlagen der Trocknungstechnik. Springer, Berlin, 1962.

[8] C. Moyne, J.C. Batsale, and A. Degiovanni. Approche expérimentale et théorique de la conductivité thermique des milieux poreux humides. II. Théorie. *Int. J. Heat Mass Transfer*, 31(11):2319-2330, 1988.

[9] I. R. Philip and D. A. De Vries. Moisture movement in porous materials under temperature gradient. Trans. Am. Geophys. Un., 38:222-232, 1957.

**Appendix: closure problem and definition of the macroscopic coefficients**

Assuming the existence of a Representative Elementary Volume, the different quantities can be computed using a so-called closure problem on a spatially periodic unit cell $Y = Y_{ha} \cup Y_i$ for the function $\boldsymbol{\chi}_\alpha$ $(\alpha \in \{ha, i\})$ with periodic boundary conditions at the external cell frontiers $Y$:

$$Y_{ha} \quad : \quad \nabla^2 \boldsymbol{\chi}_{ha} = 0 \tag{xvia}$$

$$Y_i \quad : \quad \nabla^2 \boldsymbol{\chi}_i = 0 \tag{xvib}$$

$$\partial Y_{gs} \quad : \quad \boldsymbol{\chi}_{ha} = \boldsymbol{\chi}_i \tag{xvic}$$

$$-\mathbf{n} \cdot [(k_{ha} + k_{dif})(\boldsymbol{I} + \boldsymbol{\nabla}\boldsymbol{\chi}_{ha})] = -\mathbf{n} \cdot k_i (\boldsymbol{I} + \boldsymbol{\nabla}\boldsymbol{\chi}_i) \tag{xvid}$$

Therefore the normalized water vapor diffusion coefficient is given by:

$$\mathbf{f} = \frac{\boldsymbol{D}}{\mathcal{D}_{v-a}} \quad = \quad \phi_{ha} \left( \boldsymbol{I} + \langle \boldsymbol{\nabla}\boldsymbol{\chi}_{ha} \rangle^{ha} \right) \tag{xvii}$$

where $\boldsymbol{I}$ is the identity tensor. The Fourier thermal conductivity is given by

$$\boldsymbol{k} \quad = \quad (1 - \phi_{ha}) k_i \left( \boldsymbol{I} + \langle \boldsymbol{\nabla}\boldsymbol{\chi}_i \rangle^i \right) + \phi_{ha} k_{ha} \left( \boldsymbol{I} + \langle \boldsymbol{\nabla}\boldsymbol{\chi}_{ha} \rangle^{ha} \right) \tag{xviii}$$

and the apparent thermal conductivity is given by:

$$\boldsymbol{k}_{app} \quad = \quad \boldsymbol{k} + u_{sg} \frac{\mathrm{d}\gamma_v}{\mathrm{d}\theta} \mathcal{D}_{v-a} \phi_{ha} \left( \boldsymbol{I} + \langle \boldsymbol{\nabla}\boldsymbol{\chi}_{ha} \rangle^{ha} \right) \tag{xix}$$

It is immediate to verify that the apparent thermal conductivity of the medium $\boldsymbol{k}_{app}$ is that of a two-phase medium comprising a solid phase of conductivity $k_i$ and a gas phase of conductivity $k_{ha} + k_{dif}$.

Finally note that [2]

$$\langle \boldsymbol{\nabla}\theta_{ha} \rangle^{ha} = \left( \boldsymbol{I} + \langle \boldsymbol{\nabla}\boldsymbol{\chi}_{ha} \rangle^{ha} \right) \cdot \boldsymbol{\nabla}\theta \tag{xx}$$

---

## Community Comment (CC1)

[supplement omitted: unrelated document]

---

## Author Comment (AC1)

**Response to authors' (Fourteau et al.) commentary:**

Andrew C. Hansen
22 June 2022

The authors' reply to the Comment paper affords the opportunity to bring a great deal of additional clarity to the diffusion coefficient for a layered ice/humid air microstructure.

I have provided a discussion under General Comments that succinctly sums up the two views on this topic. In brief, if one follows conventional norms for definitions of thermal conductivity and the diffusion coefficient, there is no rational debate on the subject. The diffusion coefficient for the layered microstructure is greater than that of diffusion coefficient for humid air alone. To adopt a counter point of view, supported by the authors, requires a fundamental reinterpretation of heat conduction, thermal conductivity, mass flux, and the diffusion coefficient for the mechanics of continuous media.

In addition to the general remarks provided, I have taken the time to respond in detail to all of the meaningful arguments put forth by the authors. In doing so, I have identified fundamental misconceptions of the authors' leading to identifiable errors regarding their computation of the diffusion coefficient. I also correct numerous false assertions they have made in their response.

My remarks are provided in a blue font to distinguish them from the authors commentary.

**General Comments:**

**Response to tc-2022-83**

The author of the tc-2022-83 comment argues that macroscopic water vapor diffusion flux in snow (the vapor flux at a large scale, when snow is viewed as a homogeneous medium) can be enhanced, in opposition to conclusions advanced, notably, in Fourteau et al., (2021a). Here, "enhanced" means that the macroscopic diffusion flux of vapor in snow is larger than the diffusion flux that would be observed in pure air under the same vapor concentration gradient.

The tc-2022-83 comment suffers from major flaws: it goes against physics (such as mass conservation or the difference between advection and diffusion fluxes), mixes up sources of vapor and diffusion fluxes of vapor, misunderstands what a macroscopic description is, and unfairly mischaracterizes the work it is supposed to comment on.

The opening remarks above contain no substantive information, rather, they are pure diatribe. The Comment paper does not suffer from major flaws, does not go against physics, does not confuse sources of vapor and diffusion fluxes, and finally, there is no misunderstanding of what a macroscopic description is. Any such claims to the contrary are thoroughly disabused in the detailed response provided herein.

The authors claim that the Comment paper "misunderstands what a macroscopic description is", is both naive and unfounded. I respond as someone who spent an entire academic career on multiscale phenomena and transitioning from the microscale to the macroscale. The macroscopic value of the diffusion coefficient is computed via volume averaging of a unit cell—the gold standard of the transition

from the microscale to the macroscale. Perhaps the misconception on the part of the authors stems from their improper application of volume averaging as detailed later in this response.

The authors' remark that the Comment paper "unfairly characterizes the work it is supposed to comment on" is also unfounded. Please be very specific when you make such claims. If you will kindly point out these unfair remarks, I will correct them if that is, indeed, the case. The term "unfair" should not be confused with legitimate scientific discourse.

Before proceeding with a detailed response to the authors' remarks, I provide some fundamental clarity on the differing points of view of the diffusion coefficient. These differences essentially may be reduced to fundamentally different **_definitions_** of the diffusion coefficient and how these definitions relate to heat and mass transfer in the layered microstructure of ice and humid air.

To begin, let us review several aspects of the diffusion problem leading to common ground between the authors and the Comment paper.

1.  For ice volume fractions less than 0.8, the energy flux through a layered ice/humid air microstructure may be expressed as

$$q_{lm} = -\left( \left( \frac{k_{ha}}{\phi_{ha}} \right) + \left( \frac{D_{v-a}}{\phi_{ha}} \right) u_{sg} \frac{d \, \gamma_v}{d \, \theta} \right) \frac{\partial \theta}{\partial x} \quad . \tag{A.1}$$

2.  Consider the upper and lower bounding surfaces of a layered microstructure, set between solid ice boundaries as shown in Figure A.1(a). Mass transfer across these boundaries is enhanced compared to mass transfer for humid air alone and is given by

$$\text{mass flux} = -D_{v-a} \left( \frac{d\gamma_v}{d\theta} \right) \left( \frac{\partial \theta}{\partial \xi} \right)_{ha}$$

$$= -\left( \frac{D_{v-a}}{\phi_{ha}} \right) \left( \frac{d\gamma_v}{d\theta} \right) \frac{\partial \theta}{\partial x} \quad . \tag{A.2}$$

The above is also the microscale diffusive flux in the humid air.

3.  Because of the unique layered microstructure consisting of ice and humid air layers, the energy flux of the macroscale continuum is identical to the energy flux of the ice and humid air constituents respectively, i.e.,

$$q_{lm} = q_{ha} = q_i \quad . \tag{A.3}$$

This relationship may be used to transition from the macroscale to the humid air microscale.

At this point, the approach of the authors and that presented in the Comment paper diverge with regard to macroscale definitions of conduction and diffusion. In order to proceed with a meaningful discussion of the differences in the diffusion coefficient, a control volume must be introduced.

The most natural control volume is a moving control volume coinciding with system boundaries shown by the green dashed lines in Figure A.1(b). The control volume steadily moves downward with the advancing ice front caused by ice accumulation at the upper boundary and sublimation at the lower

boundary. There are several mathematical and physical advantages of using this moving control volume including:

[Figure]

**Figure A.1.** (a) A continuum representation of a homogenized layered microstructure bounded by solid ice blocks. (b) Two control volumes for a mass transfer analysis including a fixed control volume shown by red dashed lines and a moving control volume shown by the green dashed lines.

- The mass flux transcending the upper boundary is identical to the rate of deposition of ice on the upper ice block.

- The mass flux entering through the lower boundary is identical to the rate of sublimation of mass off the lower ice block.

- The problem is steady-state, implying the configuration of the layered microstructure is constant within the control volume. Hence, there is no apparent motion of ice layers with respect to the control volume.

A consequence of the steady-state nature of the analysis is that mass flux entering through the lower boundary is identical to the mass flux leaving the upper boundary. This, in turn, implies this same mass flux is moving through the ice/humid air macroscale continuum.

At this point we can explore the different viewpoints on heat and mass transfer.

**Position A: Comment paper:**
In the case of the Comment paper, the mass flux transcending the upper or lower boundaries is used to _define_ the diffusion coefficient for the layered microstructure. The mass flux crossing the upper boundary is expressed as

$$\text{mass flux} = \gamma_v v_v = D_{lm} \left( \frac{d\gamma_v}{d\theta} \right) \frac{\partial \theta}{\partial x} \qquad , \tag{A.4}$$

where $D_{lm}$ is the diffusion coefficient of the ice/humid air macroscale mixture.

Comparing Eqs. (A.2 & A.4) reveals a diffusion coefficient defined by

$$D_{lm} = \left( \frac{D_{v-a}}{\phi_{ha}} \right) \quad . \tag{A.5}$$

Using the definition of Eq. (A.4) as the mass flux and considering the energy flux of Eq. (A.1) leads to a conductive flux given by

$$\text{conductive flux} = -k_{ha} \left( \frac{\partial \theta}{\partial \xi} \right)_{ha}$$

$$= - \left( \frac{k_{ha}}{\phi_{ha}} \right) \frac{\partial \theta}{\partial x} \quad . \tag{A.6}$$

The conductive flux leads to a traditional definition of thermal conductivity for the layered microstructure given by

$$k_{lm} = \left( \frac{k_{ha}}{\phi_{ha}} \right) \quad . \tag{A.7}$$

Equations (A.6) and (A.7) are also recognized as the classic form of conductive flux and thermal conductivity at the microscale in the *humid air of the layered microstructure*.

We emphasize that all of the above results stem from defining a diffusion coefficient of the layered microstructure that produces the mass flux crossing the upper and lower boundaries of the system as well as the mass flux through the system, all occurring due to diffusion. In the presentation that follows, we show that this definition agrees with analyses for both a moving control volume and a fixed control volume.

**Position B: Fourteau et al. (2021a):**
Fourteau et al. (2021a) introduce an alternate definition of the diffusion coefficient as the volume average (more on the volume average later) of the local diffusive flux occurring within the humid air, a seemingly natural approach. In the case of a layered microstructure, their definition of the diffusion coefficient leads to a mass flux and diffusion coefficient given by, respectively,

$$\text{mass flux} = -\phi_{ha} \left( \frac{D_{v-a}}{\phi_{ha}} \right) \left( \frac{d\gamma_v}{d\theta} \right) \frac{\partial \theta}{\partial x} \quad , \tag{A.8}$$

and

$$D_{lm} = D_{v-a} \quad . \tag{A.9}$$

Using the definition of Eq. (A.8) as the mass flux and considering the energy flux of Eq. (A.1) leads to a conductive flux and associated thermal conductivity given by

$$\text{Conductive flux} = = \left( \frac{\left( k_{ha} + \phi_i D_{v-a} \; u_{sg} \left( \frac{d\gamma_v}{d\theta} \right) \right)}{\phi_{ha}} \right) \; \frac{\partial \theta}{\partial x} \qquad , \tag{A.10}$$

and

$$k_{lm} = \left( \frac{\left( k_{ha} + \phi_i D_{v-a} \; u_{sg} \left( \frac{d\gamma_v}{d\theta} \right) \right)}{\phi_{ha}} \right) \qquad . \tag{A.11}$$

While the combination of the conductive flux and the latent heat contribution of the mass flux leads to a correct expression for the energy flux of Eq. (A.1), there are numerous troubling features related to the physics of heat and mass transfer as outlined blow.

1. The mass flux of Eq. (A.9) does produce the known mass flux transcending the upper and lower boundaries of the ice/humid air macroscale continuum. Given that the mass flux crossing the boundary is entirely due to diffusion, the authors definition of a diffusion coefficient producing diffusion less than the known mass transfer via diffusion is a fundamentally problematic result.

2. As a consequence of the steady-state nature of the moving control volume, the mass flux produced by the authors diffusion coefficient does not equal the known mass flux moving through the system which is identical to the boundary mass flux.

   The fact that the diffusion coefficient advocated by the authors does not capture the total mass flux moving through the system is readily apparent by examining the conductive flux of Eq. (A.10) *as this expression also contains a mass diffusion term.*

3. Another consequence of the authors' definition of the diffusion coefficient of Eq. (A.9) is that the thermal conductivity now includes a latent heat term as opposed to the classical definition of thermal conductivity. At first blush, including latent heat in the form of an effective thermal conductivity is not necessarily unusual. However, Eq. (A.11) includes only a portion of the latent heat, rendering an abstract physical interpretation of thermal conductivity.

In summary, the differences in the approach of the Comment paper and that of Fourteau (2021a) fall to fundamentally different **_definitions_** of the diffusion coefficient. The reader can at least make a preliminary choice of which definition to follow. A thorough mathematical analysis that follows will lay the matter to rest entirely.

We have explained in Fourteau et al. (2021a), and in a public response to a comment to Fourteau et al. (2021b) (https://tc.copernicus.org/preprints/tc-2020-317/tc-2020-317-AC1.pdf), that several fundamental problems needed to be avoided when discussing macroscopic diffusion of vapor. These problems are however at the core of the tc-2022-83 comment. Specifically, two major ones can be pointed out:

**Problem 1** (explained in 3[rd] paragraph, Section 3 of Fourteau et al., 2021a)**:**
The author of tc-2022-83 is trying to compute the macroscopic water vapor flux in an idealized layered structure of ice and air (see Figure 1 below). The combination of this

specific layered structure and of deposition/sublimation at the ice/air interface leads to a non-trivial situation, which is not treated properly in tc-2022-83. To compute the macroscopic water vapor flux, the author computes the microscopic vapor flux crossing horizontal surfaces in the air phase, between two ice layers.

However, in a layered structure of ice and air, the microscopic water vapor fluxes (and the heat conduction fluxes) differ between the ice and air phases (see Figure 1 below for the values in each phase). Computing the vapor fluxes in the air only, and ignoring what is occurring in the ice, is arbitrary and yields incorrect results.

The assertions made in the two proceeding paragraphs are absolute nonsense. Contrary to the authors' claims that I only analyzed a unit cell with boundaries through the humid air, I also analyze the case with boundaries extending through the ice phase. The results are identical in either case.

To further emphasize the role of the ice phase, I provide a mass transfer analysis herein using a fixed control volume. In doing so, I point out fundamental errors in the authors approach to computing the diffusion coefficient, errors caused by neglecting the role of the ice phase. It is ironic that their claim that the Comment paper is "ignoring what is occurring in the ice" is precisely the downfall of their own work.

**Diffusion Coefficient Analysis**

In order to determine a diffusion coefficient for the layered ice/humid air microstructure, one must begin the analysis by introducing a control volume. An excellent overview of control volumes was provided in the Comment paper (Sonin, 2001). As stated previously, the most natural control volume is a moving control volume that steadily moves downward with the advancing ice front caused by ice accumulation at the upper boundary and sublimation at the lower boundary, see Figure A.1(b). Fluid mechanics is replete with solutions involving moving control volumes and these moving control volumes often track the motion of a moving front.

Now, let us identify two different unit cells moving with the moving control volume as shown in Figure A.2. The green dashed lines are moving with the control volume, and as a consequence of the steady state nature of the problem, the respective configurations of (a) and (b) are *time independent. The static nature of the unit cells is a critical feature of the analysis that follows*.

[Figure]

**Figure A.2.** Two static configurations of a unit cell whose horizontal boundaries lie in the humid air phase, and ice phase, respectively. (a) horizontal boundary lines of the unit cell extend through the humid air, (b) horizontal boundary lines of the unit cell extend through the ice phase.

Now define a local coordinate system $(\xi)$ moving downward with the control volumes of Figure A.2. Hence, the coordinate system is moving downward at the speed of the accumulating ice front on the lower boundaries of any ice layer in the unit cell of interest.

Two interesting observations fall out:

- The problem is steady state relative to the moving reference frame, meaning the configuration of the unit cells are unchanged with time. Hence, the mass within the control volume is time independent, as is the surface flux across the boundaries of the unit cells.

- Relative to an observer on the control volume, *an arbitrary material point in the ice phase is seen moving upwards toward the upper surface of the ice* with a velocity of $v_{i/C} = -v_f$, where $v_{i/C}$ is the velocity of material point with respect to the control volume and $v_f$ is the downward moving ice front/

    A mass balance at the solid vapor interface in the unit cell yields

    $$\gamma_i v_{i/C} = -\gamma_i v_f = \gamma_v v_v \ , \tag{A.12}$$

where $v_f$ is the downward velocity of the moving ice front.

**Unit cell horizontal surfaces extending through the air phase:**
   Figure A.2(a) shows the case of the unit cell boundaries extending through the air phase. The steady state nature of the unit cell reveals:

1) The mass of the control volume is constant in time implying

    $$\frac{d}{dt}\int_{\mathcal{R}} \rho \, dV = 0 \ . \tag{A.13}$$

The Reynold's Transport Theorem for mass conservation may be expressed in the form (Sonin, 2001)

$$\frac{d}{dt}\int_{\mathcal{R}(t)}\rho\,dV + \int_{\partial\mathcal{R}(t)}\gamma_v(v_v - v_C)\,dV = 0 \quad, \tag{A.14}$$

where $v_C$ is the control surface velocity. Recognizing $v_C = v_f$, the transport theorem for water vapor reduces to

$$\int_{\partial\mathcal{R}(t)}\gamma_v(v_v - v_f)\,dV = 0 \quad. \tag{A.15}$$

Noting $(v_f / v_v)$ is on the order of $10^{-6}$, there follows

$$\int_{\partial\mathcal{R}(t)}\gamma_v v_v\,dV = 0 \quad. \tag{A.16}$$

The above simply implies the mass of water vapor entering the control volume from below is equal to the mass of water vapor leaving the control volume from above.

2) The control volume boundaries continuously lie at the interface between the solid ice and the ice mixture. Hence, mass transfer across the control volume surface is precisely the mass flux of water vapor crossing the boundaries between the humid air and the bounding solid ice. The mass flux across the upper and lower boundaries is governed by

$$\text{mass flux} = \gamma_v v_v = \left(\frac{D_{v-a}}{\phi_{ha}}\right)\left(\frac{d\gamma_v}{d\theta}\right)\frac{\partial\theta}{\partial x} \quad. \tag{A.17}$$

The authors have falsely claimed that the Comment paper is based solely on this analysis, a claim refuted directly below.

**Unit cell horizontal surfaces extending through the ice phase:**
Consider the unit cell of Figure A.2(b) where the unit cell boundaries extend through the ice phase. Again, the unit cell remains steady-state (time independent) implying:

1. The mass of the control volume is constant in time, i.e.,

$$\frac{d}{dt}\int_{\mathcal{R}}\rho\,dV = 0 \quad. \tag{A.18}$$

The Reynold's Transport Theorem for mass conservation may be expressed in the form (Sonin, 2001)

$$\frac{d}{dt}\int_{\mathcal{R}(t)}\rho\,dV + \int_{\partial\mathcal{R}(t)}\gamma_i(v_i - v_C)\,dV = 0 \quad, \tag{A.19}$$

where $v_C$ is the control surface velocity. The transport theorem through the ice phase reduces to

$$\int_{\partial\mathcal{R}(t)}\gamma_i v_{i/C}\,dV = 0 \quad. \tag{A.20}$$

Noting Eq. (A.12) leads to an expression for *the mass flux across ice boundaries that is identical mass flux across humid air boundaries*, i.e.,

$$\text{mass flux} = \gamma_v v_v = \left(\frac{D_{v-a}}{\phi_{ha}}\right)\left(\frac{d\gamma_v}{d\theta}\right)\frac{\partial\theta}{\partial x} \qquad . \tag{A.21}$$

**Volume averaging the mass flux:**

Using either unit cell from Figure A.2, the volume average of the mass flux over the entire volume of the unit cell is given by

$$\text{mass flux} = \quad \phi_{ha}\,\gamma_v\,v_v + \phi_i\gamma_i v_{i/C}$$

$$= (\phi_{ha} + \phi_i)\,\gamma_v\,v_v = \gamma_v\,v_v = \left(\frac{D_{v-a}}{\phi_{ha}}\right)\left(\frac{d\gamma_v}{d\theta}\right)\frac{\partial\theta}{\partial x} \qquad . \tag{A.22}$$

Note that *the volume averaged mass flux and the surface mass flux agree.* This feature is important, as the authors incorrectly criticize the Comment paper for using the surface flux to compute the macroscale diffusion coefficient. To alleviate this complaint, use the volume average as shown in Eq. (A.22) to compute the macroscale diffusion coefficient, the conclusions are unchanged.

The authors claim that I have neglected the role of the ice phase, i.e. "computing the vapor fluxes in the air only, and ignoring what is occurring in the ice, is arbitrary and yields incorrect results." This statement is so unhinged from the work in this Comment paper it defies a logical response and suggests the authors did not seek to truly understand the development.

The entire Comment paper is devoted to the role the ice phase plays in macroscale diffusion. In addition to the mathematical analysis above showing the role of the ice phase, let me provide some additional language from the Comment paper describing the important role of the ice phase.

> **L532:534:** ***Perhaps the most important aspect of the unit cell analysis of the moving control volume is that the ice phase is a contributing factor to the overall mass transport of water moving through the system.*** As a result, diffusion of water vapor is enhanced at all humid air volume fractions.

> **L839-848:** This comment paper demonstrates that hand-to-hand water vapor transport provides an effective model for correctly predicting enhanced diffusion in a layered ice/humid air microstructure. The model is supported by rigorous control volume analyses using both a moving control volume and a fixed control volume. Although the hand-to-hand concept is incredibly valuable, one can dispense with the hand-to-diffusion mechanism and still achieve the same results of enhanced diffusion due to the "reservoir effect" of the ice phase holding massive amounts of water vapor. In brief, ***the existing ice phase within the layered microstructure is a major contributing factor to the overall mass transport of water moving through the system.*** The approach of Fourteau et al. (2021a) ignores the contribution of mass diffusion attributed to the reservoirs of water vapor contained within the ice layers.

In summary, the Comment paper is devoted to the contributing factor of the ice phase in the role of the macroscopic diffusion coefficient. I would add that the bold and italicized print in the above was written in the original submission of the Comment paper to emphasize the importance of this point. It is bewildering as to how the authors could make claims to the exact contrary of what I expressly wrote and developed mathematically. I don't believe the above statements taken from the Comment paper could be any clearer.

More on the role of the ice phase and the authors' failure to account for this role is presented in an analysis of a fixed control volume later in this reply.

This is shown with a simple reduction to absurdity (a powerful word that is profoundly incorrectly used, given the authors' misunderstanding): with the same reasoning, one could compute the water vapor fluxes crossing a horizontal surface in the ice phase (the ice being as much a part of snow as the air), and find that the macroscopic vapor flux is null. Thus, the same reasoning leads to different diffusion coefficients for the same snow microstructure, in contradiction with the notion that a given snow microstructure has a unique macroscopic description. In other words, trying to obtain the macroscopic flux as the flux through given horizontal surfaces is flawed.

The authors have a fundamental misconception here. In a moving control volume, there is a mass flux of ice relative to the control volume. As the reader will see, it is the authors' failure to clearly define whether a control volume is moving or fixed that leads to their errors in developing the diffusion coefficient.

What happens is that by focusing on just the air phase, the author is misinterpreting local microscopic fluctuations as the actual macroscopic description. At the macroscopic scale, these microscopic fluctuations average out into the volume-average. The removal of these fluctuations in effective transport coefficients is one of the basis of a macroscopic description (e.g. Quintard and Whitaker, 1990). Computing the vapor fluxes in the air only, and ignoring what is occurring in the ice, is arbitrary and yields incorrect results.

Let us be crystal clear here regarding the Comment paper:

- The macroscale diffusion coefficient may be computed directly from volume averaging in the moving control volume, owing to the steady-state nature of the problem.

- The volume averaged diffusion coefficient is identical to the surface flux.

The authors' comments stating the Comment paper only uses unit cell boundaries that lie in the humid air as well as determining the mass flux through horizontal surfaces of the unit cell is a familiar refrain that occurs ad nauseum throughout the remainder of their review. Their claims in these matters are categorically false.

[Figure]

*Figure 1:* Mass, heat conduction, and total energy fluxes in a layered structure. The demonstrations in the tc-2022-83 comment essentially revolve around computing the fluxes in the air phase only (where vapor flux is maximal and conduction flux minimal), neglecting the fact that these fluxes are different in the ice.

Figure 1 above strikes at the very heart of the matter and reveals the fundamental error in the authors' approach of volume averaging to compute the diffusion coefficient. As noted previously, one cannot properly address the heat and mass transfer of the layered ice/humid air microstructure without a control volume surrounding the material. It is the point of departure for any integral formulation of a mass transfer analysis in elementary fluid mechanics. A lack of a clearly defined control volume will set you adrift.

While the authors never explicitly state they are using a fixed control volume in Figure 1, their calculations show implicitly that this is the case. In what follows, I identify the precise error made by the authors' in attempting to compute a diffusion coefficient using a fixed control volume.

Let figure A.3(a) represent a fixed control volume containing a single layer of ice and a single layer of humid air. Figure A.3(b) also represents the *same* fixed control volume at a later time, showing an updated location of the ice phase. The time dependent nature of the ice phase location within the unit cell precludes the possibility of treating the fixed control volume as a unit cell. As a result, the simple volume average of the microscale diffusive flux does not compute the true mass transfer.

For the control volume of Figure A.3, begin by noting the microscale diffusive flux in the humid air is given by

$$\text{microscale mass flux} = v_{\mathrm{v}}\, \gamma_{\mathrm{v}} = -D_{\mathrm{v-a}} \left(\frac{d\gamma_{\mathrm{v}}}{d\theta}\right) \left(\frac{\partial \theta}{\partial \xi}\right)_{\mathrm{ha}}$$

$$= -\left(\frac{D_{\mathrm{v-a}}}{\phi_{\mathrm{ha}}}\right) \left(\frac{d\gamma_{\mathrm{v}}}{d\theta}\right) \frac{\partial \theta}{\partial x} \quad . \tag{A.23}$$

[Figure]

**Figure A.3.** Fixed control volume (a) and the same control volume at a later time (b) showing the location of the ice phase after complete mass turnover of an ice layer. Note the relative positions of points $P$ and $Q$ *that move entirely due to diffusion.* Let the length of the control volume have dimension 1, such that volume fractions also represent lineal fractions.

*NOTE*: It is important to clearly understand the differences in control volumes of Figures A.2 and A.3. Figure A.2 shows two entirely different control volumes whose configurations (location of the ice phase within the control volume) are static for all time. In contrast, Figure A.3 shows the same control volume at two different times. In other words, the location of the ice phase in the control volume of Figure A.3 is time dependent.

Now consider two material points of water, $P$ and $Q$, residing in the ice phase as shown in Figure A.3(a). These points move to points $P'$ and $Q'$, respectively, showing the ice phase in Figure A.3(a) has turned over entirely and moved to the location shown in Figure A.3(b). We make two critical observations regarding mass transfer:

1.  The entire mass in the ice phase located in Figure A.3(a) has moved to the location in Figure A.3(b) via diffusion. This allows one to compute a mass flux by determining the time it takes for this location change of the ice phase to occur.

2.  For complete mass turnover of the ice phase, the location of the ice phase has relocated by a distance, $L_{ha} = \phi_{ha}$. This is a critical observation and one that is not recognized by the authors' volume averaging scheme—the total diffusion path is shortened compared to the length of the control volume.

The time, $\tau$, required to move from the ice location from Figure A.3(a) to Figure A.3(b) is identical to the time taken for point $Q$ to reach $Q'$ and is given by

$$\tau = \frac{\phi_i}{v_f} + \frac{\phi_{ha}}{v_v}. \tag{A.24}$$

The first time in the above equation represents the "wait time" while point $Q$ is locked in the ice phase, waiting to reach humid air via sublimation of ice above. The second term represents the "travel time" via diffusion to arrive at $Q'$. Note that the second term is on the order of $10^{-6}$ of the first term and may be neglected. Now use Eq. (A.12) to express the velocity of the moving front, $v_f$, as

$$v_{\mathrm{f}} = \left(\frac{\gamma_{\mathrm{v}}}{\gamma_{\mathrm{i}}}\right) v_{\mathrm{v}} \tag{A.25}$$

Substituting the above into Eq. (A.24) and neglecting the negligible term $\left(\frac{\phi_{\mathrm{ha}}}{v_{\mathrm{v}}}\right)$ yields

$$\tau = \frac{\phi_{\mathrm{i}}\,\gamma_{\mathrm{i}}}{v_{\mathrm{v}}\,\gamma_{\mathrm{v}}} = \frac{m}{v_{\mathrm{v}}\,\gamma_{\mathrm{v}}} \tag{A.26}$$

where $m$ is the mass of the ice layer. Rearranging the above shows that mass flux of ice moving through the system is

$$v_{\mathrm{v}}\,\gamma_{\mathrm{v}} = \frac{m}{\tau}$$

Or noting Eq. (A.23) the mass flux of ice moving through the system via diffusion is identically

$$v_{\mathrm{v}}\,\gamma_{\mathrm{v}} = -\left(\frac{D_{\mathrm{v-a}}}{\phi_{\mathrm{ha}}}\right)\left(\frac{d\gamma_{\mathrm{v}}}{d\theta}\right)\frac{\partial\theta}{\partial x}\quad. \tag{A.27}$$

The elevated mass transfer rate is readily observed in the sequence shown in Figure A.4, showing: (a) an initial time, (b) the configuration after 1 turnover of an ice layer, and (c) the configuration after 2 turnovers of the ice phase. The figure very clearly depicts the elevated mass transfer described by the fixed control volume analysis above. *Notice that the distance point P moves during ice turnover is $L_{\mathrm{ha}} = \phi_{\mathrm{ha}}$. This phenomenon will occur if there are 2 layers or two thousand layers. Mass transfer is elevated and the diffusion coefficient is given by*

$$\boxed{D_{\mathrm{lm}} = \left(\frac{D_{\mathrm{v-a}}}{\phi_{\mathrm{ha}}}\right)}\quad.$$

The fact that the mass turnover moves the ice location by $L_{\mathrm{ha}} = \phi_{\mathrm{ha}}$ and not the entire distance across the control volume given by $L = 1$ is the cause of the elevated macroscale diffusion in the ice/humid air mixture. The authors' simplistic volume averaging does not account for this very subtle feature of mass transfer in the layered microstructure. In brief, they have not properly assed the role of the ice phase in a fixed control volume analysis.

[Figure]

**Figure A.4.** Sequence of ice phase turnover showing: (a) an initial time, (b) the configuration after 1 turnover of an ice layer, and (c) the configuration after 2 turnovers of the ice phase. Notice the distance traveled by a point $P$ is shorter than the total distance across a fixed control volume. This is the so called "reservoir effect" referenced in the Comment paper.

**Analysis Summary**

As a summary of "Problem 1" identified by the authors, the results of the preceding diffusion coefficient analysis afford one the opportunity to make a multifaceted collection of observations and conclusions:

- The diffusion coefficient for the layered ice humid air mixture is enhanced compared to the diffusion coefficient of humid air alone. Moreover, the diffusion coefficient yields a mass flux that is in complete agreement with the known mass flux crossing the upper and lower boundaries via diffusion as well as the mass flux moving through the system.

- The analysis of the unit cell for the moving control volume was completed with unit cell boundaries extending through humid air as well as boundaries extending through the ice phase. The results in either case are identical. Hence, the authors' claim that I chose boundaries through humid air to achieve a desired outcome is unequivocally false.

- The diffusion coefficient may be computed by volume averaging the mass flux over the entire unit cell of a moving control volume—the gold standard approach to homogenization of microscale field variables to achieve a macroscopic field variable. The authors claim that the paper "misunderstands what a macroscopic description is" is unequivocally false.

- We have now established that the diffusion coefficient is enhanced. The analysis of Section 5.1, tracking a material point, $A$, of water, shows that the motion of point $A$ through the system is due entirely to diffusion through humid air. Moreover, as point $A$ moves through the system, it never goes over, though, or around an ice layer—*there is no hand-to-hand mass transport*! In summary:

- o Mass transfer in the humid air microstructure is enhanced compared to mass transfer in humid air alone,

- o Mass transfer is due entirely to diffusion with no hand-to-hand mass transport.

- I have identified the fundamental error made by the authors in computing a diffusion coefficient via volume averaging of a fixed control volume. Specifically, they have neglected the consequences of movement of the location of the ice phase within the control volume. When this error is corrected, one arrives at the enhanced diffusion coefficient presented in the Comment paper. *Hence, using either a moving control volume or a fixed control volume, the diffusion coefficient is the same, yielding an enhanced value of diffusion compared to diffusion through humid air alone.*

**Problem 2** (explained in 2ⁿᵈ and 3ʳᵈ paragraphs, Section 2 of Fourteau et al., 2021a)**:**

As noted above, the Comment paper does not rely on the concept of hand-to-hand diffusion to develop the diffusion coefficient. As a consequence, the entire discussion of "Problem 2" is completely irrelevant and need not be addressed.

The authors appear to be attempting to address the work of Hansen (2019). That paper was devoted to analyzing the diffusion coefficient using Yosida's (1955) famous analogy of hand-to-hand mass transport. I'll simply note the diffusion coefficient results of Hansen (2019) are identical to what is computed here without the use of the hand-to-hand diffusion analogy.

> **L373-377:** While the above discussion provides a cogent physical explanation of the role of hand-to-hand vapor transport in the diffusion coefficient, one may argue that the discussion lacks the necessary mathematical rigor to be wholly defensible. This weakness is dispelled in Sections 4 & 5 through a rigorous control volume analysis, as well as tracking a material point of water throughout its life in traveling through the microstructure to the upper solid ice boundary.

In summary, the hand-to-hand concept of diffusion is convenient to produce the correct diffusion coefficient for the ice/humid air microstructure. However, to avoid the arguments of this mechanism entirely, the concept of the diffusion coefficient was developed in the Comment paper without the use of the hand-to-hand analogy. This is yet another example of the authors failing to acknowledge what is clearly written in the paper while moving on to a false narrative that is simply not relevant. The text below, bounded by asterisks, has zero bearing on this Comment paper.
* * *
To support enhanced macroscopic water vapor fluxes, the tc-2022-83 comment is also invoking the "hand-to-hand" mechanism, in which the simultaneous deposition of a water molecule below an ice grain and sublimation at the top of the ice grain is counted as an actual mass transport, as if the water vapor molecule were instantaneously transferred across the ice.

However, the simultaneous deposition and sublimation of vapor simply does not produce any mass transport across the ice. No mass is spatially transported through the ice in the process, as molecules stay where they are during sublimation/deposition (see Figure 2a below and Figure 1 of Fourteau et al., 2021a).

The tc-2022-83 comment also points to the indistinguishability of water vapor molecules as supporting the hand-to-hand mechanism. The argument says that since water molecules are indistinguishable, everything really appears as if the vapor molecule instantaneously moved across the ice. However, indistinguishability does not change the situation: no mass is transferred during sublimation/deposition. Indeed, if one argues that it appears as if the vapor molecule teleported through the ice phase, then the same argument would also hold for the water molecule in the ice phase: the molecule at the top surface would re-appear at the bottom surface, canceling the hypothetical hand-to-hand flux of vapor across the ice. All in all, no net mass is spatially transferred during the simultaneous deposition/sublimation (see Figure 2b). The better (and simpler) physical description is of course not to invoke these apparent fluxes and the indistinguishability of molecules, and to only consider diffusion fluxes in the pores, as done in Fourteau et al., (2021a).

Finally, by arguing that the situation is equivalent to one where the depositing vapor molecule would have been instantaneously transferred through the ice, two physics principles are violated: (i) matter does not teleport and (ii) mass cannot simply disappear (1 molecule is missing in Figure 2c).

[Figure]

***Figure 2:*** *Three visions on what happens during the deposition/sublimation of vapor. a) The first one is the actual/real one (and there is no mass transfer). b) The second is what can be said by properly using the argument of indistinguishability (it is a weird view of things, but it is consistent with the actual physics). c) The third one is the hand-to-hand argument: it violates mass conservation and the notion that infinitely fast mass transport is impossible.*

\*\*\*\*\*\*\*\*\*\*\*\*\*\*\*\*\*\*\*\*\*\*\*\*\*\*\*\*\*\*\*\*\*\*\*\*\*\*\*\*\*\*\*\*\*\*\*\*\*\*\*\*\*\*\*\*\*\*\*\*\*\*\*\*\*\*\*\*\*\*\*\*\*\*\*\*\*\*\*\*\*\*\*\*

The tc-2022-83 comment essentially revolves around Problem 1: the mass and heat conduction fluxes are computed only in the air phase through various methods (from direct computation, from mass balance in a control volume having its surface boundaries in the air, from deposition at the ice/air interface, etc), and then wrongly attributed to the macroscopic scale, as if results obtained in the air applied to the whole structure. Problem

2 is also regularly invoked as a supporting interpretation of the results yielded by Problem 1.

*The above remarks of the authors are repetitive, tiresome, and incorrect. There is absolutely zero truth to any of them as has been demonstrated in detail in the diffusion coefficient analysis above.*

To support its conclusion, the tc-2022-83 comment also introduced the "reservoir effect": there is a vast amount of water molecules in the ice waiting to sublimate and diffuse in the air. However, this whole argument is based on a confusion between vapor sources (that do not move mass across space) and vapor diffusion fluxes (that actually move mass from one place to another). It is a variation of Problem 2.

*The above paragraph is patently false. There is no "confusion between vapor sources (that do not move mass across space) and vapor diffusion fluxes (that actually move mass from one place to another)."*

*Section 5 very clearly demonstrates that all mass transfer occurs by water vapor diffusing through humid air. That said, let us focus on a specific layer of ice and track the motion of material points of water. At the upper surface of a layer, water vapor is sublimating off the ice phase and subsequently diffusing through humid air until it reaches the next ice layer. Over time, the entire ice phase within an ice layer diffuses through humid air as the material points of water sublimate off the upper surface, are released into the humid air, and diffuse through the humid air (see the fixed control volume analysis for graphic confirmation). In brief, the ice phase acts as a source of water vapor, all of which diffuses through the humid air and contributes to the macroscale diffusion coefficient.*

*Now consider a material point A of water vapor located within the humid air phase in the middle of the macroscale continuum. As point A proceeds upward during diffusion, note that the water contained in every ice layer above point A reaches the upper boundary before point A reaches the upper boundary. In other words, point A never passes around through or over an ice layer—there is no hand-to-hand diffusion mechanism!*

*The ice layers above point A clearly act as an enormous source of water vapor that moves through the macroscale continuum via diffusion. The authors claim that I have confused a source with transport via diffusion is unequivocally false.*

Apart from that, we also want to stress that the tc-2022-83 comment mischaracterizes what is actually said in Fourteau et al. (2021a) (e.g. Specific Comments L16 and L25 below) and does not reply to the points raised in it (e.g. L364-372 below). Scientific standards would expect the author to properly describe the article they are commenting on. Failure to do so is detrimental to readers and referees, and deteriorates the quality of the scientific debate.

*I have addressed the authors specific remarks in L16 and L25 below. The comments regarding scientific standards and the detriment to readers and referees are simply unprofessional. They contain no substantive information, have no bearing on the paper, and represent an unwarranted personal attack. The Comment paper is a thoughtful response to authors' paper, filled with rigorous arguments that are thoroughly defensible.*

Finally, we note that the tc-2022-83 comment contradicts 30 years of well-established scientific literature on the subject of the macroscopic description of humid materials. For instance, Whitaker (1998) provides a demonstration that the macroscopic vapor diffusion flux is the volume average of the microscopic fluxes (right-hand-side of Equation 162). Section 5.1 of Moyne et al. (1988) provides an analysis of the exact layered problem that the tc-2022-83 comment is focusing on. The normalized diffusion coefficient is given in Equation 60 and the thermal conductivity in Equation 59. They are both equal to the expressions given in Appendix C of Fourteau et al. (2021a). Moyne et al. (1991) provides a general expression of the normalized diffusion coefficient based on the averaged temperature gradient in the air phase (Equation 12). This is consistent with what we obtained in Equation 5 of Fourteau et al. (2021a) and the exact same expression as in Equation 10 of Fourteau et al. (2021b).

The authors have misrepresented the historical evidence by only citing select literature that supports their view, as if that is the entire story. Here are some examples that counter their view:

de Quervain, M.R.: On the metamorphosis of snow. In W.D. Kingery (Ed.), Ice and Snow (pp. 377-390). Cambridge, MA:  The M.I.T. Press., 1963.

Colbeck, S.: The vapor diffusion coefficient for snow, Water Resour. Res., 29, 109–115, https://doi.org/10.1029/92WR02301, 1993.

Sommerfeld, R., Friedman, I., and Nilles, M.: The fractionation of natural isotopes during temperature gradient metamorphism of snow, in: The Fractionation of Natural Isotopes During Temperature Gradient Metamorphism of Snow, 95–105, D. Reidel Publishing, Boston, 1987.

Christon, M., Burns, P., and Sommerfeld, R.: Quasi-steady temperature gradient metamorphism in idealized, dry snow, Numer. Heat Tr. A-Appl., 25, 259–278, https://doi.org/10.1080/10407789408955948, 1994.

Hansen, A. C.: Revisiting the vapor diffusion coefficient in dry snow, Preprint. Discussion, The Cryosphere, https://doi.org/10.5194/tc-2019-143, 2020.

I think there is a much finer point to put on the historical analysis. The Comment paper should not be viewed as a paper siding with past history, as if adding weight to a scale of justice. On the contrary, the paper is about the natural evolutionary advance of science. It is a notable contribution to the understanding of diffusion in ice/humid air mixtures.

By ignoring well-established techniques of homogenization, the tc-2022-83 represents a step back in the understanding of water vapor movement in porous media.

Actually, it is the authors who have incorrectly performed a homogenization of the macroscale diffusion coefficient. There is no debate here. Furthermore, the error I have exposed in the authors' work represents a fundamental contribution to understanding mass transfer in ice/humid air mixtures, including snow.

The authors of Fourteau et al. (2021a)

**Specific Comments:**

Here, we quickly list some of the tc-2022-83 comment's biggest flaws , as an illustration of what was summarized in the General Comments.

**L16:** Claiming that we consider that *"the ice phase has no effect other than occupying volume where diffusion cannot occur"* is false. We actually spent most of our article discussing the influence of the coupling between the diffusion flux and the presence of ice through surface effects

Let us be a bit more specific about the authors' approach and their failure to account for the role of the ice phase. As described previously, the volume averaging performed by the authors using the fixed control volume is problematic in that there is no true unit cell for a fixed control volume. A unit cell must be geometrically repeating (true here) with a configuration that is time independent (not true here). There are only two appropriate paths toward computing the diffusion coefficient correctly using control volumes. Either:

1. use a moving control volume to achieve a steady-state configuration of a unit cell and appropriately volume average, or

2. use a fixed control volume while computing the mass transfer as the result of configuration changes in the location of the ice phase within the control volume.

**L25:** We never argued that there is an actual *"counterflux of water vapor in the form of downward motion of ice"* and are well aware that this downward motion is only an appearance (as exemplified by the sentence *"for an observer focused on the ice everything **appears** as if the ice disappearing on the sublimation side reappeared on the depositing side"*).

A quote of text taken directly from Fourteau (2021a; pg 391) reads:" Because of mass conservation during the sublimation and deposition process, the apparent flux of vapor skipping the ice is compensated by an equal counter-flux of water molecules in the ice space."

To be crystal clear, there is no counterflux of mass. All material points of water move upward in a monotonically increasing (nonnegative) fashion. The apparent downward motion of ice is actually the result of upward motion of water vapor from below.

We actually argued that the author's "hand-to-hand" mass flux is also only apparent, and that to be consistent they should consider both apparent fluxes and not just the one that suits their conclusion (*"the **apparent** flux of vapor skipping the ice is compensated by an equal counter-flux of water molecules in the ice phase"*). See Figure 2 of this response.

I don't believe the authors ever referred to the counterflux as an "apparent counterflux." How is one supposed to interpret what is written differently?

**L42:** The releasing of water vapor from the ice phase is a source, not a transport flux. It does not spatially move mass, it only changes its form (from ice to vapor). There is no mass transport other than when and where vapor diffuses in the pore.

The ice phase does, indeed, act as an enormous source of water vapor and that water vapor, when released from the ice via sublimation, and transported via diffusion through the humid air. Where is the controversy here? I explained all of this in detail previously in this reply.

**L79:** The indistinguishability of water molecules does not create a mass transport (Problem 2).

At the risk of yet another repetitive response, none of the key results of this Comment paper ever invoke hand-to-hand diffusion analogy and the indistinguishability of water molecules. Again, it appears the authors wish to argue with Hansen (2019) which, although correct, is an entirely different discussion from the present Comment paper.

**L109:** Of course, gradient metamorphism can exist without molecules teleporting through the ice, and the author should be aware of that as it is explained in Fourteau et al. (2021a). We clearly stated at the end of Section 2 of Fourteau et al. (2021a) that one should be careful to separate between the notions that (i) water molecules preferentially deposit below ice grains under temperature gradient (which is physically sound and visible in the simulations in Figures 3 and 5 of Fourteau et al., 2021a), and (ii) that such deposition patterns act as mass transport (which is Problem 2).

The Comment paper never argues that molecules are "teleported" through the ice, rather the notion of hand-to-hand diffusion is a useful analogy. Let us stay on point: water vapor passing through a layered microstructure moves entirely due to diffusion through humid air pores and that diffusion produces an enhanced mass flux compared to the mass flux of humid air alone.

**L170:** Problem 1. The author computes the diffusive mass flux in the air phase (and the deposition/sublimation flux at the ice/air interface, which has an equal magnitude), but then inappropriately interprets it as the macroscopic diffusive flux.

Let me again disabuse the authors of this familiar complaint. The macroscopic diffusive flux is computed correctly using volume averaging in the moving control volume. Moreover, the volume average over the entire volume is identical to the mass flux through humid air alone at the microscale.

In contrast, the volume averaging suggested by the authors, using the implicitly defined fixed control volume, is invalid, owing to the time dependence of the location of the ice phase within the control volume. A unit cell does not exist relative to the time domain. A proper accounting of the mass flux caused by the ice phase location changes produces the macroscale diffusion coefficient that is identical to the moving control volume analysis—physics demands this to be true.

**L258:** Problem 1. The author computes the diffusive flux in the air, but then wrongly interprets it as the macroscopic diffusive flux.

A casual review of the authors comments to this point shows at least 5 occurrences with variations of the above statement. The statement, for at least the fifth time, is not true. Furthermore, the continual claims of wrongly interpreting the macroscopic diffusive flux are also unequivocally false.

**L299-310:** It is normal that Eqs 16 and 17 do not reproduce the mass and conduction fluxes in the air, since they are supposed to give the macroscopic flux (that would be Problem 1 otherwise).

I have no idea what the authors are attempting to articulate here. That said, the macroscopic energy flux is identical to the energy flux through the humid air, per Eq. (A.3).

**L335-353:** Term (2) simply corresponds to the extra heat conduction in the ice phase (see the orange curve in Figure 1 of the response), scaled by the amount of ice that carries this extra conduction flux.

As this extra heat conduction in the ice phase accommodates the release/absorption of latent heat at the ice/air interface (in order to respect the continuity of the total energy flux at the ice/air interface), it is perfectly normal that it depends on latent heat and diffusion of vapor.

The Comment paper produces a diffusive flux that accounts for all mass diffusion moving through the macroscale ice/humid air mixture and a conductive flux that is consistent with a diffusing, or nondiffusing, mixture and supported by the kinetic theory of gases. The paper of Fourteau (2021a) yields both a diffusive flux and a conductive flux in abstract terms that are simply indefensible.

**L364-372:** Problem 2 (sublimation/deposition does not create a mass transfer). This whole discussion is also unnecessary when one realizes that term (2) is simply the extra conduction flux in the ice due to microscopic latent heat/conduction coupling (see Section 2.2 and Section 3.1 of Fourteau et al., 2021b for a discussion on this physical effect).

I believe the correct approach is to either exclude diffusion from thermal conductivity (preferred) or include all terms of diffusion into an effective thermal conductivity. The approach of Fourteau (2021a) does neither—it is simply an abstract set of new definitions for heat and mass transfer, forced on them by an incorrect application of volume averaging the diffusion coefficient using a fixed control volume.

The author should be aware of that, as that is exactly what we pointed out in Section 4.3 of Fourteau et al. (2021a) when discussing Hansen and Folsien (2015): *"However, during the identification of the latent-heat contribution to the total energy flux, **some of the heat conduction contribution of the ice is attributed to the latent-heat transport**".*

Interesting comment but a more accurate statement of the authors' work would read "***some of the heat conduction contribution of the ice is attributed to 'some of' the latent-heat transport***". In other words, the authors' definition of thermal conductivity only includes *some* of the latent heat. This, again, is an artifact of incorrectly defining the diffusion coefficient.

**Section 4.2:** The author is trying to compute the macroscopic advecto-diffusive (with an advection and a diffusion component) water flux within a moving control volume. The control surfaces are placed in the air phase. (This statement is not true—control surfaces were modeled in both the air phase and the ice phase. The drumbeat of this false narrative continues.)

- Eq. 37 is false, the microscopic advection flux of vapor at the boundaries is missing ( $v_{i/c}\gamma_v$ ).

The authors claim of the missing term identified as $(v_{i/c}\gamma_v)$ is true! That said, this term is effectively zero, by any measure, compared to the remaining term.

The exact water vapor crossing a boundary is

$$\int_{\partial\mathcal{R}(t)} \gamma_v(v_v - v_f)\, dV = 0 \ . \tag{30}$$

where $v_{i/C} = -v_f$

Let us now reference the following text from the Comment paper:

> **L422-424:** A subtle but important observation throughout the analysis of this section is that the true water vapor diffusion velocity is unaffected by the speed of the advancing ice front as $(\hat{v}_f / \hat{v}_v)$ is on the order of $10^{-6}$.

The hats on the velocity terms above apply to humid air as a pure substance but the argument for the layered microstructure is identical. Hence, the term the authors claim I have neglected is on the order of $10^{-6}$ of the remaining term—effectively zero by any engineering or scientific perspective.

- First line of Eq. 38 is false, the averaged advection flux of vapor is missing ( $\varphi_{ha}v_{i/c}\gamma_v$ ).

- With the advection of vapor properly included, the volume averaged advecto-diffusive flux does not equal the advecto-diffusive flux on the air surface boundaries. They differ by $\varphi_i v_{i/c}\gamma_v$ .

The advection vapor term is negligible (zero!) per the above discussion. This is nothing more than sound scientific judgement, I dropped a term that is of order $10^{-6}$ of the term that remains.

This is normal since the advection and diffusion fluxes in the air are not representative of the whole microstructure (variation of Problem 1).

- This (wrongly) computed macroscopic advecto-diffusive water flux is finally inappropriately interpreted as the sole macroscopic diffusive flux.

While it may be a bit tiresome to the reader, none of these remarks are a legitimate concern. The term "wrongly", used on occasion here is rather amusing. If I throw a wrench across a room and use rigid body dynamics to compute its trajectory, have I "wrongly" computed a result because I neglected the spin of the earth which I know exists? Of course, this is a nonsensical discussion, one that is in the same vein as what the authors are suggesting for the diffusion problem.

To sumup Section 4.2: The macroscopic advecto-diffusion flux is wrongly computed by dropping off the advection of vapor. Then, the advection of ice is lumped with the diffusion of vapor and the ensemble is interpreted as a pure diffusion flux (neglecting the fact that

extra advective water fluxes have been introduced through the motion of the reference frame).

The authors seem to be suggesting that the extra "advective" water flux introduced through the motion of the reference frame somehow calls into question the analysis? I hope that is not the case.

When one wants to understand a specific problem in mechanics, we are free to choose a control volume, either moving with a specified velocity or fixed. The choice of control volume is strictly done to compute the values in the problem of interest. In the case of mass transfer through a layered ice/humid air microstructure, by far, the most natural control volume is one that tracks the moving front caused by ice accumulation at the upper surface and ice depletion via sublimation at the lower surface. Moreover, the configuration of the layers is static implying the mass flux through the system is the same as mass transfer crossing the boundaries. Would the authors care to refute the physical correctness of the moving control volume?

For those (perhaps the authors are in this camp) who insist on using a fixed control volume, the complete accounting of water vapor diffusion must address the changing configuration within the control volume and the net effect on the mass flux through the system. This concept was thoroughly developed in this reply, with the results producing an enhanced diffusion coefficient consistent with that predicted using a moving control volume.

To try a simple analogy, this is akin to saying that someone can run at 50km/hr because they are moving at this velocity in the reference frame of a moving car. To obtain the actual speed of the runner respective to the ground, we need to retract the advection term due to the movement of the reference frame. The same thing applies to obtain the diffusion flux of water in a moving reference frame.

Let's expand on the above analogy with a bit more precision using terms with the correct relative orders of magnitude. Let us consider a flat rail car where I walk at 1 m/s. However, the rail car is moving, albeit *very* slowly at a speed of $10^{-6}$ m/s. How fast would you say I am walking? Of course, the answer is 1 m/s from any rational engineering analysis. In brief, I could walk 1000 km in the time the freight car moves 1 m! By any definition, I am walking at 1 m/s and the motion of the rail car is negligible. This is the same argument I have invoked for the diffusion case. The authors wish to retain a term that is of order $10^{-6}$ of the term retained.

If the author includes the advection of vapor in Eq. 38, and then removes the macroscopic advective flux of water ($v_{i/c}<\gamma_{water}> = v_{i/c}(\varphi_{ha}\gamma_v + \varphi_i\gamma_i)$), they would find that the remaining diffusive flux is given by the volume-averaged microscopic diffusion fluxes (as in Moyne et al., 1991, Whitaker et al., 1998, or Fourteau et al., 2021a).

Why would one remove the advective flux of water in the form of the ice phase? There is no mathematical or physical justification for such a leap.

Instead of arguing this point with the authors, let us revert to fundamental fluid mechanics. Choose a control volume (fixed or moving) and correctly apply all the principles of mass transfer. The result is an enhanced mass flux compared to the mass flux of humid air alone and the entire mass flux is due to diffusion—full stop.

**Section 4.3:** There is no conflict to resolve here: microscopic vapor fluxes depend on whether we are in the air or the ice. The difference between the two is accommodated through sublimation/deposition at the ice/air interface. If the lower and upper microscopic boundary fluxes differ, it means that there is a net accumulation/depletion in the control volume that accommodates the microscopic flux imbalance.

Section 4.3 is rigorously correct. It is yet another *independent* approach to showing diffusion enhancement compared to diffusion through humid air alone.

But sublimation and deposition are vapor sources/sinks, not diffusion fluxes of vapor. They do not transfer mass in space (Problem 2). (Again, the Problem 2 discussion is not relevant.) The vapor that is released at the ice/air interface was not spatially moved during the process.

It has never been suggested that vapor sources/sinks are related to diffusive flux—they are simply a source/sink of vapor. I will simply repost the following text:

There is no "confusion between vapor sources (that do not move mass across space) and vapor diffusion fluxes (that actually move mass from one place to another)." Section 5 very clearly demonstrates that all vapor diffusion occurs by water vapor moving through humid air. That said, let us focus on a specific layer of ice and track the motion of material points of water. At the upper surface, water vapor is sublimating off the ice phase and subsequently diffusing through humid air until it reaches the next ice layer. Over time the entire ice phase within an ice layer diffuses through humid air as the material points of water sublimate off the upper surface, are released into the humid air and diffuses through the humid air. In brief, the ice phase acts as a source of water vapor, all of which diffuses through the humid air and contributes to the macroscale diffusion coefficient. There is no "confusion between vapor sources (that do not move mass across space) and vapor diffusion fluxes (that actually move mass from one place to another)."

The reservoir effect is just a confusion between sources (that do not move mass) and diffusion fluxes (that do move mass).

A demonstrably false assertion!

**L699-706:** The lake analogy is wrong and at the heart of Problem 1. There is no local sources and accumulation/depletion of mass in a steady state lake: thus what goes in must go out, in every control volume. That is why one can find that all the microscopic mass fluxes integrated across a perpendicular surface spanning the lake are equal: all surfaces have a representative mass flux . In this peculiar case, the "macroscopic" mass flux going through the lake can be obtained by simply considering the inflow or outflow fluxes.

I agree with everything the authors have stated above. For steady-state conditions, mass flow into the lake equals mass flow out of the lake equals mass flow moving through the lake.

In the layered ice structure, there is local accumulation/depletion of mass (at the ice/air interface): what goes in can accumulate and stay there. Thus, the flux is not constant through any perpendicular surfaces (see Figure 1 of the response), and one cannot simply look at mass fluxes through a simple surface (Problem 1).

The claim above is also demonstrably false when analyzing the problem using the moving control volume. In this case, identical to the lake, mass flow into the control volume equals mass flow out of the control volume equals mass moving through the control volume. In the humid air, the mass flow is due to diffusion. In the ice, the mass transfer is a direct result of the moving control volume while the ice is stationary. Hence, ice is moving upward with respect to the control volume. The analogy with the lake is precise in this regard.

**L714-720:** The author got it correct that point A just stay in the ice phase, potentially being released later, where it will start diffusing again from the very same point in space where it deposited. However, this incorporation of water in ice does not create any mass flux across the ice phase at the moment of deposition (this would be Problem 2), and there is no mass movement besides when and where vapor diffuses in the air.

The authors repeated reference to Problem 2 is misleading to the reader—the discussion is not relevant. I never refer to "mass flux across the ice phase" in the analysis of the moving control volume or the fixed control volume.

Let us focus on the material point of water described in Section 5.1. All mass movement occurs where vapor diffuses through humid air just as the authors suggest above. In brief, water vapor deposits on the lower side of an ice layer, resides there until all ice in the layer is removed via sublimation, and then continues on via vapor diffusion. There is no reference to hand-to-hand diffusion in this process!

**L723-L734:** The author is arguing that since a molecule in the ice might be eventually released, from the exact same place where it deposited and after sufficient time has elapsed, this molecule should be counted has having skipped across the ice phase during its deposition. This is absurd.

To begin, all material points of water that condense on an ice layer (in a 1D analysis) will eventually be released at what will now be the upper surface after all ice above it sublimates away. The authors use the phrase "might be eventually released" as if this "might" not happen. I really don't know how to respond to this notion other than to say *it will happen and we know exactly when it will happen. This is a classic example of ice layer turnover. Further, all motion of any material point of water is caused by water vapor diffusion.*

The latter phrase in the authors' comment states "this molecule should be counted has having skipped across the ice phase during its deposition. This is absurd." This is a ***wild statement*** that has no merit whatsoever and certainly is not applicable to the analysis I presented. Nowhere in Section 5.1 have I suggested that the molecule should be counted as having skipped across the ice phase. In fact, the exact opposite is what I have asserted—there is no jumping over or passing through ice as the molecule resides in the ice phase until it reaches the upper surface and then continues its march via diffusion in humid air. Where is skipping the ice mentioned in the discussion of Section 5.1?

**Section 5.1:** This section is a convoluted way of falling into Problem 1.

First, the author computes the mass deposited below the ice during a time interval $\tau$ as the product of the deposition rate and $\tau$ (Eq. 48). They then find that this quantity of mass is larger than the deposition that would occur if the deposition flux were equal to the diffusion flux in pure air.

This result is trivially stating that the deposition flux in the layered structure is larger than the diffusion flux in pure air under a similar gradient, something we already knew and agreed upon (see Figure 1 of this response). But interpreting the elevated deposition flux as an elevated macroscopic flux is Problem 1 (confusion between local microscopic fluxes and the macroscopic one).

*Do the authors really want to suggest that the elevated deposition flux caused by diffusion is somehow different than the mass flux due to diffusion resulting from their definition of the diffusion coefficient? I already demonstrated the specific error in their diffusion coefficient calculation. The authors are simply in an indefensible position. They have made a fatal error in their volume averaging analysis of a fixed control volume—such a control volume does not represent a unit cell as the configuration is time dependent*

*Finally, once again, I have no confusion between local microscopic fluxes and the macroscopic one. This point is thoroughly addressed on multiple occasions in this reply.*

It might also seem that the demonstration relies on tracking mass movement over time, since the chosen time interval is the time it would take a "material point" to travel from a sublimation surface to a deposition one. However, the demonstration would be the same whatever the time interval $\tau$ (this can easily be seen by keeping m as $\gamma_v v_v \tau$ in the computation). Thus, the notion of tracking mass movement during a deposition/sublimation cycle is irrelevant to the demonstration.

*The authors are correct in that the demonstration in Section 5.1 relies on tracking mass movement over time. There is certainly nothing wrong with such an approach. One can compute mass moving through a system instantaneously or over any period of time, just the same as one could track heat moving through a system instantaneously or over a period of time.*

*The authors are also quite correct that the demonstration is the same, no matter the time interval. Precisely! The time interval is irrelevant and the results are the same, mass transfer via diffusion is elevated compared to diffusion in humid air alone.*

**L888:** Most of the errors in the tc-2022-83 comment originate from the confusion between the macroscopic flux and local microscopic fluxes through horizontal surfaces, which strongly differ in the case of the layered microstructure.

*The authors statement here could not be more underlined{profoundly incorrect}. It is simply an outlandish claim—one that is repeatedly made throughout this commentary. In fact, the opposite is true, it is the authors who have misconceptions as they have not fully accounted for the enhanced diffusion effects caused by the location of the ice phase moving within a fixed control volume.*

However, for random heterogeneous media, such as snow, this problem disappears, as microstructural fluctuations across horizontal layers tend to statistically average out. In this case the mass fluxes through horizontal surfaces are equal to the volume average (as explained in our public response https://tc.copernicus.org/preprints/tc-2020-317/tc-2020-317-AC1.pdf).

The problems I have identified for the layered microstructure do not disappear in random heterogeneous media such as snow. In the case of small temperature gradients, it is quite possible that the majority of diffusion occurs by moving through the pore spaces with little condensation or sublimation. However, as temperature gradients rise, the path of water vapor will become more linear, aligned with the temperature gradient, resulting in mass transfer similar to the layered microstructure considered herein. Quantifying the effects of temperature gradient and other microstructural variables would be a wonderful area of research.

We took the time to compute the mass fluxes crossing the various horizontal surfaces of a snow microstructure with a finite-element simulation (Figure 3 below): the surface fluxes are basically all the same (with small microscale fluctuations). They are below the flux that would be observed in the air under a similar macroscopic vapor gradient (i.e. not enhanced).

This statement is, of course true, but does nothing to diminish the errors made by the authors regarding water vapor diffusion in a layered microstructure. These same errors will appear in snow under strong temperature gradients.

So, not only is the idea of computing fluxes through individual horizontal layers giving strongly erroneous results in the case of the layered structure, it would not even lead to enhanced vapor fluxes in the case of real snow.

This statement is not true for the same reasons articulated in this Comment paper.

[Figure]

*Figure 3:* *Vapor fluxes across different horizontal surfaces in a snow sample (in blue). They slightly fluctuate around the average (in orange), and are inferior to the flux in pure air under a similar gradient (black). With a larger sample, the fluctuation would decrease further.*

**Conclusion**

Let us begin by identifying two points of agreement:

1. The mass flux is enhanced in the ice/humid air mixture compared to the mass flux in humid air alone.

2. The mass flux is due entirely to water vapor diffusion. Furthermore, there is no hand to hand mass transfer.

And now for the differences.

1.   The Comment paper defines a diffusion coefficient that captures the entire diffusive mass transfer moving through the mixture and crossing system boundaries. This value is computed independently, among other ways, using a fixed control volume, a moving control volume, and tracking a material point of water.

2.   The authors define a diffusion coefficient that produces a mass flux that is less than the known mass flux moving through the system via diffusion. This result is caused by an incorrect application of volume averaging of a fixed control volume.

In closing, despite the persistent fierce nature of the authors' critique of the Comment paper, they are unable to refute a single aspect of the paper. The paper provides notable contributions to the understanding of ice/humid air mixtures and is technically on *terra firma*. In contrast, the authors have introduced a diffusion coefficient that does not capture the known mass transfer due to diffusion—that is simply an untenable position.

Lastly, the research on the layered microstructure has broad implications to mass transfer in snow in that high temperature gradients will lead to linear diffusion paths that produce mass transfer mechanisms similar to a layered microstructure

**References:**

Calonne, N., Geindreau, C., and Flin, F.: Macroscopic modeling for heat and water vapor transport in dry snow by homogenization, J. Phys. Chem. B, 118, 13393-13403, https://doi.org/10.10221/jp5052535, 2014.

K. Fourteau, F. Domine, and P. Hagenmuller: Macroscopic water vapor diffusion is not enhanced in snow, The Cryosphere, 15, 389–406, doi:10.5194/tc-15-389-2021, 2021a

K. Fourteau, F. Domine, and P. Hagenmuller: Impact of water vapor diffusion and latent heat on the effective thermal conductivity of snow, The Cryosphere, 15, 2739–2755, doi:10.5194/tc-15-2739-2021, 2021b

A.C. Hansen and W.E. Foslien: A macroscale mixture theory analysis of deposition and sublimation rates during heat and mass transfer in dry snow, The Cryosphere, 9, 1857–1878, doi:10.5194/tc-9-1857-2015, 2015

C. Moyne, J.C. Batsale and A. Degiovanni: Approche expérimentale et théorique de la conductivité thermique des milieux poreux humides – II. Théorie, Int. J. Heat Mass Transfer., 31, 11, 2319-2330, 1988, doi:10.1016/0017- 9310(88)90163-9

C. Moyne and P. Perre: Processes related to drying: Part I, Theoretical model. Drying Tech., 9, 5, 1135-1152, 1991, doi:10.1080/07373939108916746

M. Quintard and S. Whitaker: Two-phase flow in heterogeneous porous media I: The influence of large spatial and temporal gradients, Transp. Porous Media, 5, 341-379, 1990

Sonin, A.A.: Fundamental laws of motion for particles, material volumes, and control volumes, MIT, https://web.mit.edu/2.25/www/pdf/fundamental_laws.pdf, 2001.

Yosida, Z.: Physical studies of deposited snow: thermal properties I, Tech. rep., Institute of low temperature science, Hokkaido University, Japan, 1955.

S. Whitaker: Coupled transport in multiphase systems: A theory of drying, Adv. Heat Transfer., 31, 1998, url:

https://www.researchgate.net/publication/241065896_Coupled_Transport_in_Multiphase_Systems_A_Theory_of_Drying (last access: 03/06/2022)